# SymmetricDiffusers: Learning Discrete Diffusion on Finite Symmetric Groups

**Yongxing Zhang**[1,3*]**, Donglin Yang**[2,3]**, Renjie Liao**[2,3]
[1]University of Waterloo    [2]University of British Columbia    [3]Vector Institute
`nick.zhang@uwaterloo.ca, {ydlin, rjliao}@ece.ubc.ca`

## Abstract

The group of permutations $S_n$, also known as the finite symmetric groups, are essential in fields such as combinatorics, physics, and chemistry. However, learning a probability distribution over $S_n$ poses significant challenges due to its intractable size and discrete nature. In this paper, we introduce *SymmetricDiffusers*, a novel discrete diffusion model that simplifies the task of learning a complicated distribution over $S_n$ by decomposing it into learning simpler transitions of the reverse diffusion using deep neural networks. We identify the riffle shuffle as an effective forward transition and provide empirical guidelines for selecting the diffusion length based on the theory of random walks on finite groups. Additionally, we propose a generalized Plackett-Luce (PL) distribution for the reverse transition, which is provably more expressive than the PL distribution. We further introduce a theoretically grounded "denoising schedule" to improve sampling and learning efficiency. Extensive experiments show that our model achieves state-of-the-art or comparable performance on solving tasks including sorting 4-digit MNIST images, jigsaw puzzles, and traveling salesman problems. Our code is released at `https://github.com/DSL-Lab/SymmetricDiffusers`.

## 1 Introduction

As a vital area of abstract algebra, finite groups provide a structured framework for analyzing symmetries and transformations which are fundamental to a wide range of fields, including combinatorics, physics, chemistry, and computer science. One of the most important finite groups is the *finite symmetric group $S_n$*, defined as the group whose elements are all the bijections (or permutations) from a set of $n$ elements to itself, with the group operation being function composition.

Classic probabilistic models for finite symmetric groups $S_n$, such as the Plackett-Luce (PL) model (Plackett, 1975; Luce, 1959), the Mallows model (Mallows, 1957), and card shuffling methods (Diaconis, 1988), are crucial in analyzing preference data and understanding the convergence of random walks. Therefore, studying probabilistic models over $S_n$ through the lens of modern machine learning is both natural and beneficial. This problem is theoretically intriguing as it bridges abstract algebra and machine learning. For instance, Cayley's Theorem, a fundamental result in abstract algebra, states that every group is isomorphic to a subgroup of a symmetric group. This implies that learning a probability distribution over finite symmetric groups could, in principle, yield a distribution over any finite group. Moreover, exploring this problem could lead to the development of advanced models capable of addressing tasks such as permutations in ranking problems, sequence alignment in bioinformatics, and sorting.

However, learning a probability distribution over finite symmetric groups $S_n$ poses significant challenges. First, the number of permutations of $n$ objects grows factorially with $n$, making the inference and learning computationally expensive for large $n$. Second, the discrete nature of the data brings difficulties in designing expressive parameterizations and impedes the gradient-based learning.

In this work, we propose a novel discrete-time discrete (state space) diffusion model over finite symmetric groups, dubbed as *SymmetricDiffusers*. It overcomes the above challenges by decomposing the difficult problem of learning a complicated distribution over $S_n$ into a sequence of simpler

---

*Work done while an intern at Vector Institute.

problems, *i.e.*, learning individual transitions of a reverse diffusion process using deep neural networks. Based on the theory of random walks on finite groups, we investigate various shuffling methods as the forward process and identify the riffle shuffle as the most effective. We also provide empirical guidelines on choosing the diffusion length based on the mixing time of the riffle shuffle. Furthermore, we examine potential transitions for the reverse diffusion, such as inverse shuffling methods and the PL distribution, and introduce a novel generalized PL distribution. We prove that our generalized PL is more expressive than the PL distribution. Additionally, we propose a theoretically grounded "denoising schedule" that merges reverse steps to improve the efficiency of sampling and learning. To validate the effectiveness of our SymmetricDiffusers, we conduct extensive experiments on three tasks: sorting 4-Digit MNIST images, solving Jigsaw Puzzles on the Noisy MNIST and CIFAR-10 datasets, and addressing traveling salesman problems (TSPs). Our model achieves the state-of-the-art or comparable performance across all tasks.

## 2 RELATED WORKS

**Random Walks on Finite Groups.** The field of random walks on finite groups, especially finite symmetric groups, have been extensively studied by previous mathematicians (Reeds, 1981; Gilbert, 1955; Bayer & Diaconis, 1992; Saloff-Coste, 2004). Techniques from a variety of different fields, including probability, combinatorics, and representation theory, have been used to study random walks on finite groups (Saloff-Coste, 2004). In particular, random walks on finite symmetric groups are first studied in the application of card shuffling, with many profound theoretical results of shuffling established. A famous result in the field shows that 7 riffle shuffles are enough to mix up a deck of 52 cards (Bayer & Diaconis, 1992), where a riffle shuffle is a mathematically precise model that simulates how people shuffle cards in real life. The idea of shuffling to mix up a deck of cards aligns naturally with the idea of diffusion, and we seek to fuse the modern techniques of diffusion models with the classical theories of random walks on finite groups.

**Diffusion Models.** Diffusion models (Sohl-Dickstein et al., 2015; Song & Ermon, 2020; Ho et al., 2020; Song et al., 2021) are a powerful class of generative models that typically deals with continuous data. They consist of forward and reverse processes. The forward process is typically a discrete-time continuous-state Markov chain or a continuous-time continuous-state Markov process that gradually adds noise to data, and the reverse process learn neural networks to denoise. Discrete (state space) diffusion models have also been proposed to handle discrete data like image, text (Austin et al., 2023), and graphs (Vignac et al., 2023). However, existing discrete diffusion models focused on cases where the state space is small or has a special (*e.g.*, decomposable) structure and are unable to deal with intractable-sized state spaces like the symmetric group. In particular, Austin et al. (2023) requires an explicit transition matrix, which has size $n! \times n!$ in the case of finite symmetric groups and has no simple representations or sparsifications. Finally, other recent advancement includes efficient discrete transitions for sequences (Varma et al., 2024), continuous-time discrete-state diffusion models (Campbell et al., 2022; Sun et al., 2023; Shi et al., 2024) and discrete score matching models (Meng et al., 2023; Lou et al., 2024), but the nature of symmetric groups again makes it non-trivial to adapt to these existing frameworks.

**Differentiable Sorting and Learning Permutations.** A popular paradigm to learn permutations is through differentiable sorting or matching algorithms. Various differentiable sorting algorithms have been proposed that uses continuous relaxations of permutation matrices (Grover et al., 2018; Cuturi et al., 2019; Blondel et al., 2020), or uses differentiable swap functions (Petersen et al., 2021; 2022; Kim et al., 2024). The Gumbel-Sinkhorn method (Mena et al., 2018) has also been proposed to learn latent permutations using the continuous Sinkhorn operator. Such methods often focus on finding the optimal permutation instead of learning a distribution over the finite symmetric group. Moreover, they tend to be less effective as $n$ grows larger due to their high complexities.

## 3 LEARNING DIFFUSION MODELS ON FINITE SYMMETRIC GROUPS

We first introduce some notations. Fix $n \in \mathbb{N}$. Let $[n]$ denote the set $\{1, 2, \ldots, n\}$. A *permutation* $\sigma$ on $[n]$ is a function from $[n]$ to $[n]$, and we usually write $\sigma$ as $\begin{pmatrix} 1 & 2 & \cdots & n \\ \sigma(1) & \sigma(2) & \cdots & \sigma(n) \end{pmatrix}$. The *identity permutation*, denoted by Id, is the permutation given by $\mathrm{Id}(i) = i$ for all $i \in [n]$. Let $S_n$ be the set of all permutations (or bijections) from a set of $n$ elements to itself, called the *finite*

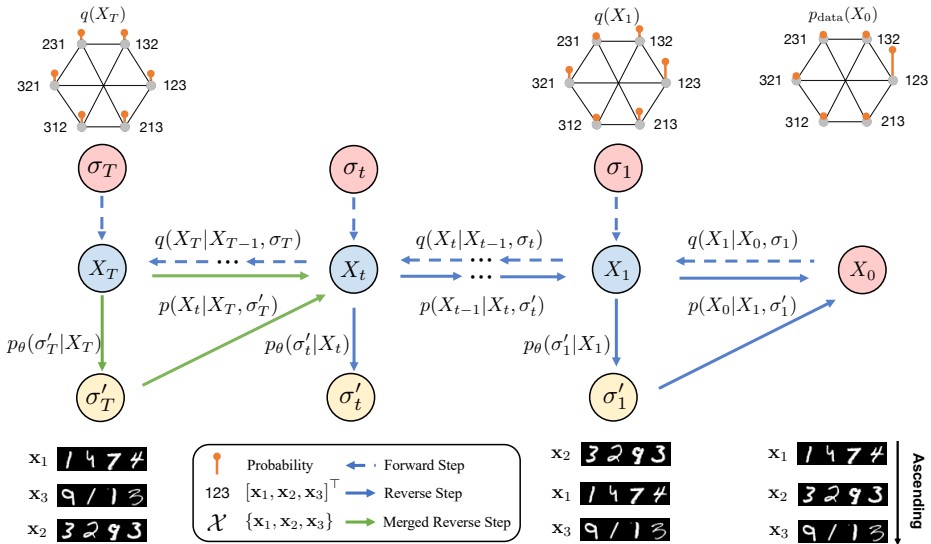

Figure 1: This figure illustrates our discrete diffusion model on finite symmetric groups. The middle graphical model displays the forward and reverse diffusion processes. We demonstrate learning distributions over the symmetric group $S_3$ via the task of sorting three MNIST 4-digit images. The top part of the figure shows the marginal distribution of a ranked list of images $X_t$ at time $t$, while the bottom shows a randomly drawn list of images.

*symmetric group*, whose group operation is the function composition. For a permutation $\sigma \in S_n$, the permutation matrix $Q_\sigma \in \mathbb{R}^{n \times n}$ associated with $\sigma$ satisfies $e_i^\top Q_\sigma = e_{\sigma(i)}^\top$ for all $i \in [n]$. In this paper, we consider a set of $n$ distinctive objects $\mathcal{X} = \{\mathbf{x}_1, \ldots, \mathbf{x}_n\}$, where the $i$-th object is represented by a $d$-dimensional vector $\mathbf{x}_i$. Therefore, a ranked list of objects can be represented as a matrix $X = [\mathbf{x}_1, \ldots, \mathbf{x}_n]^\top \in \mathbb{R}^{n \times d}$, where the ordering of rows corresponds to the ordering of objects. We can permute $X$ via permutation $\sigma$ to obtain $Q_\sigma X$.

Our goal is to learn a distribution over $S_n$. We propose learning discrete (state space) diffusion models, which consist of a *forward process* and a *reverse process*. In the forward process, starting from the unknown data distribution, we simulate a random walk until it reaches a known stationary "noise" distribution. In the reverse process, starting from the known noise distribution, we simulate another random walk, where the transition probability is computed using a neural network, until it recovers the data distribution. Learning a transition distribution over $S_n$ is often more manageable than learning the original distribution because: (1) the support size (the number of states that can be reached in one transition) could be much smaller than $n!$, and (2) the distance between the initial and target distributions is smaller. By doing so, we break down the hard problem (learning the original distribution) into a sequence of simpler subproblems (learning the transition distribution). The overall framework is illustrated in Fig. 1. In the following, we will introduce the forward card shuffling process in Section 3.1, the reverse process in Section 3.2, the network architecture and training in Section 3.3, denoising schedule in Section 3.4, and reverse decoding methods in Section 3.5.

### 3.1 FORWARD DIFFUSION PROCESS: CARD SHUFFLING

Suppose we observe a set of objects $\mathcal{X}$ and their ranked list $X_0$. They are assumed to be generated from an unknown data distribution in an IID manner, *i.e.*, $X_0, \mathcal{X} \overset{\text{iid}}{\sim} p_{\text{data}}(X, \mathcal{X})$. One can construct a bijection between a ranked list of $n$ objects and an ordered deck of $n$ cards. Therefore, permuting objects is equivalent to shuffling cards. In the forward diffusion process, we would like to add "random noise" to the rank list so that it reaches to some known stationary distribution like the uniform. Formally, we let $\mathcal{S} \subseteq S_n$ be a set of permutations that are realizable by a given shuffling method in one step. $\mathcal{S}$ does not change across steps in common shuffling methods. We will provide concrete examples later. We then define the *forward process* as a Markov chain,

$$q(X_{1:T}|X_0, \mathcal{X}) = q(X_{1:T}|X_0) = \prod_{t=1}^{T} q(X_t|X_{t-1}), \qquad (1)$$

where $q(X_t|X_{t-1}) = \sum_{\sigma_t \in \mathcal{S}} q(X_t|X_{t-1}, \sigma_t)q(\sigma_t)$ and the first equality in Eq. (1) holds since $X_0$ implies $\mathcal{X}$. In the forward process, although the set $\mathcal{X}$ does not change, the rank list of objects $X_t$ changes. Here $q(\sigma_t)$ has the support $\mathcal{S}$ and describes the permutation generated by the underlying

shuffling method. Note that common shuffling methods are time-homogeneous Markov chains, *i.e.*, $q(\sigma_t)$ stays the same across time. $q(X_t|X_{t-1}, \sigma_t)$ is a delta distribution $\delta\left(X_t = Q_{\sigma_t} X_{t-1}\right)$ since the permuted objects $X_t$ are uniquely determined given the permutation $\sigma_t$ and $X_{t-1}$. We denote the *neighbouring states* of $X$ via one-step shuffling as $N_{\mathcal{S}}(X) := \{Q_\sigma X | \sigma \in \mathcal{S}\}$. Therefore, we have,

$$q(X_t|X_{t-1}) = \begin{cases} q(\sigma_t) & \text{if } X_t \in N_{\mathcal{S}}(X_{t-1}) \\ 0 & \text{otherwise.} \end{cases} \tag{2}$$

Note that $X_t \in N_{\mathcal{S}}(X_{t-1})$ is equivalent to $\sigma_t \in \mathcal{S}$ and $X_t = Q_{\sigma_t} X_{t-1}$.

### 3.1.1 CARD SHUFFLING METHODS

We now consider several popular shuffling methods as the forward transition, *i.e.*, *random transpositions*, *random insertions*, and *riffle shuffles*. Different shuffling methods provide different design choices of $q(\sigma_t)$, thus corresponding to different forward diffusion processes. Although all these forward diffusion processes share the same stationary distribution, *i.e.*, the uniform, they differ in their mixing time. We will introduce stronger quantitative results on their mixing time later.

**Random Transpositions.** One natural way of shuffling is to swap pairs of objects. Formally, a *transposition* or a *swap* is a permutation $\sigma \in S_n$ such that there exist $i \neq j \in [n]$ with $\sigma(i) = j$, $\sigma(j) = i$, and $\sigma(k) = k$ for all $k \notin \{i, j\}$, in which case we denote $\sigma = (i \quad j)$. We let $\mathcal{S} = \{(i \quad j) : i \neq j \in [n]\} \cup \{\text{Id}\}$. For any time $t$, we define $q(\sigma_t)$ by choosing two indices from $[n]$ uniformly and independently and swap the two indices. If the two chosen indices are the same, then this means that we have sampled the identity permutation. Specifically, $q(\sigma_t = (i \quad j)) = 2/n^2$ when $i \neq j$ and $q(\sigma_t = \text{Id}) = 1/n$.

**Random Insertions.** Another shuffling method is to insert the last piece to somewhere in the middle. Let $\texttt{insert}_i$ denote the permutation that inserts the last piece right before the $i^{\text{th}}$ piece, and let $\mathcal{S} := \{\texttt{insert}_i : i \in [n]\}$. Note that $\texttt{insert}_n = \text{Id}$. Specifically, we have $q(\sigma_t = \texttt{insert}_i) = 1/n$ when $i \neq n$ and $q(\sigma_t = \text{Id}) = 1/n$.

**Riffle Shuffles.** Finally, we introduce the riffle shuffle, a method similar to how serious card players shuffle cards. The process begins by roughly cutting the deck into two halves and then interleaving the two halves together. A formal mathematical model of the riffle shuffle, known as the *GSR model*, was introduced by Gilbert and Shannon (Gilbert, 1955), and independently by Reeds (1981). The model is described as follows. A deck of $n$ cards is cut into two piles according to binomial distribution, where the probability of having $k$ cards in the top pile is $\binom{n}{k}/2^n$ for $0 \leq k \leq n$. The top pile is held in the left hand and the bottom pile in the right hand. The two piles are then riffled together such that, if there are $A$ cards left in the left hand and $B$ cards in the right hand, the probability that the next card drops from the left is $A/(A + B)$, and from right is $B/(A + B)$. We implement the riffle shuffles according to the GSR model. For simplicity, we will omit the term "GSR" when referring to riffle shuffles hereafter.

There exists an exact formula for the probability over $S_n$ obtained through one-step riffle shuffle. Let $\sigma \in S_n$. A *rising sequence* of $\sigma$ is a subsequence of $\sigma$ constructed by finding a maximal subset of indices $i_1 < i_2 < \cdots < i_j$ such that permuted values are contiguously increasing, *i.e.*, $\sigma(i_2) - \sigma(i_1) = \sigma(i_3) - \sigma(i_2) = \cdots = \sigma(i_j) - \sigma(i_{j-1}) = 1$. For example, the permutation $\begin{pmatrix} 1 & 2 & 3 & 4 & 5 \\ 1 & 4 & 2 & 5 & 3 \end{pmatrix}$ has 2 rising sequences, *i.e.*, 123 (red) and 45 (blue). Note that a permutation has 1 rising sequence if and only if it is the identity permutation. Denoting by $q_{\text{RS}}(\sigma)$ the probability of obtaining $\sigma$ through one-step riffle shuffle, it was shown by Bayer & Diaconis (1992) that

$$q_{\text{RS}}(\sigma) = \frac{1}{2^n}\binom{n+2-r}{n} = \begin{cases} (n+1)/2^n & \text{if } \sigma = \text{Id} \\ 1/2^n & \text{if } \sigma \text{ has two rising sequences} \\ 0 & \text{otherwise,} \end{cases} \tag{3}$$

where $r$ is the number of rising sequences of $\sigma$. The support $\mathcal{S}$ is thus the set of all permutations with at most two rising sequences. We let the forward process be $q(\sigma_t) = q_{\text{RS}}(\sigma_t)$ for all $t$.

### 3.1.2 MIXING TIMES AND CUT-OFF PHENOMENON

All of the above shuffling methods have the uniform distribution as the stationary distribution. However, they have different mixing times (*i.e.*, the time until the Markov chain is close to its

stationary distribution measured by some distance), and there exist quantitative results on their mixing times. Let $q \in \{q_{\text{RT}}, q_{\text{RI}}, q_{\text{RS}}\}$, and for $t \in \mathbb{N}$, let $q^{(t)}$ be the marginal distribution of the Markov chain after $t$ shuffles. We describe the mixing time in terms of the total variation (TV) distance between two probability distributions, *i.e.*, $D_{\text{TV}}(q^{(t)}, u)$, where $u$ is the uniform distribution.

For all three shuffling methods, there exists a *cut-off phenomenon*, where $D_{\text{TV}}(q^{(t)}, u)$ stays around 1 for initial steps and then abruptly drops to values that are close to 0. The *cut-off time* is the time when the abrupt change happens. For the formal definition, we refer the readers to Definition 3.3 of Saloff-Coste (2004). In Saloff-Coste (2004), they also provided the cut-off time for random transposition, random insertion, and riffle shuffle, which are $\frac{n}{2} \log n$, $n \log n$, and $\frac{3}{2} \log_2 n$ respectively. Observe that the riffle shuffle reaches the cut-off much faster than the other two methods, which means it has a much faster mixing time. Therefore, we use the riffle shuffle in the forward process.

## 3.2 THE REVERSE DIFFUSION PROCESS

We now model the *reverse process* as another Markov chain conditioned on the set of objects $\mathcal{X}$. We denote the set of realizable *reverse permutations* as $\mathcal{T}$, and the neighbours of $X$ with respect to $\mathcal{T}$ as $N_{\mathcal{T}}(X) := \{Q_\sigma X : \sigma \in \mathcal{T}\}$. The conditional joint distribution is given by

$$p_\theta(X_{0:T}|\mathcal{X}) = p(X_T|\mathcal{X}) \prod_{t=1}^{T} p_\theta(X_{t-1}|X_t), \tag{4}$$

where $p_\theta(X_{t-1}|X_t) = \sum_{\sigma_t' \in \mathcal{T}} p(X_{t-1}|X_t, \sigma_t') p_\theta(\sigma_t'|X_t)$. To sample from $p(X_T|\mathcal{X})$, one simply samples a random permutation from the uniform distribution and then shuffle the objects accordingly to obtain $X_T$. $p(X_{t-1}|X_t, \sigma_t')$ is again a delta distribution $\delta(X_{t-1} = Q_{\sigma_t'} X_t)$. We have

$$p_\theta(X_{t-1}|X_t) = \begin{cases} p_\theta(\sigma_t'|X_t) & \text{if } X_{t-1} \in N_{\mathcal{T}}(X_t) \\ 0 & \text{otherwise,} \end{cases} \tag{5}$$

where $X_{t-1} \in N_{\mathcal{T}}(X_t)$ is equivalent to $\sigma_t' \in \mathcal{T}$ and $X_{t-1} = Q_{\sigma_t'} X_t$. In the following, we will introduce the specific design choices of the distribution $p_\theta(\sigma_t'|X_t)$.

### 3.2.1 INVERSE CARD SHUFFLING

A natural choice is to use the inverse operations of the aforementioned card shuffling operations in the forward process. Specifically, for the forward shuffling $\mathcal{S}$, we introduce their inverse operations $\mathcal{T} := \{\sigma^{-1} : \sigma \in \mathcal{S}\}$, from which we can parameterize $p_\theta(\sigma_t'|X_t)$.

**Inverse Transposition.** Since the inverse of a transposition is also a transposition, we can let $\mathcal{T} := \mathcal{S} = \{(i \quad j) : i \neq j \in [n]\} \cup \{\text{Id}\}$. We define a distribution of inverse transposition (IT) over $\mathcal{T}$ using $n + 1$ real-valued parameters $\mathbf{s} = (s_1, \ldots, s_n)$ and $\tau$ such that

$$p_{\text{IT}}(\sigma) = \begin{cases} 1 - \phi(\tau) & \text{if } \sigma = \text{Id}, \\ \phi(\tau) \left( \dfrac{\exp(s_i)}{\sum\limits_k \exp(s_k)} \cdot \dfrac{\exp(s_j)}{\sum\limits_{k \neq i} \exp(s_k)} + \dfrac{\exp(s_j)}{\sum\limits_k \exp(s_k)} \cdot \dfrac{\exp(s_i)}{\sum\limits_{k \neq j} \exp(s_k)} \right) & \text{if } \sigma = \begin{pmatrix} i & j \end{pmatrix}, i \neq j, \end{cases} \tag{6}$$

where $\phi(\cdot)$ is the sigmoid function. The intuition behind this parameterization is to first handle the identity permutation Id separately, where we use $\phi(\tau)$ to denote the probability of not selecting Id. Afterwards, probabilities are assigned to the transpositions. A transposition is essentially an *unordered* pair of *distinct* indices, so we use $n$ parameters $\mathbf{s} = (s_1, \ldots, s_n)$ to represent the logits of each index getting picked. The term in parentheses represents the probability of selecting the unordered pair $i$ and $j$, which is equal to the probability of first picking $i$ and then $j$, plus the probability of first picking $j$ and then $i$.

**Inverse Insertion.** For the random insertion, the inverse operation is to insert some piece to the end. Let $\texttt{inverse\_insert}_i$ denote the permutation that moves the $i^{\text{th}}$ component to the end, and let $\mathcal{T} := \{\texttt{inverse\_insert}_i : i \in [n]\}$. We define a categorical distribution of inverse insertion (II) over $\mathcal{T}$ using parameters $\mathbf{s} = (s_1, \ldots, s_n)$ such that,

$$p_{\text{II}}(\sigma = \texttt{inverse\_insert}_i) = \exp(s_i) / \left( \sum_{j=1}^{n} \exp(s_j) \right). \tag{7}$$

**Inverse Riffle Shuffle.** In the riffle shuffle, the deck of card is first cut into two piles, and the two piles are riffled together. So to undo a riffle shuffle, we need to figure out which pile each card belongs to, *i.e.*, making a sequence of $n$ binary decisions. We define the Inverse Riffle Shuffle (IRS) distribution using parameters $\mathbf{s} = (s_1, \ldots, s_n)$ as follows. Starting from the last (the $n^{\text{th}}$) object, each object $i$ has probability $\phi(s_i)$ of being put on the top of the left pile. Otherwise, it falls on the top of the right pile. Finally, put the left pile on top of the right pile, which gives the shuffled result.

### 3.2.2 THE PLACKETT-LUCE DISTRIBUTION AND ITS GENERALIZATION

Other than specific inverse shuffling methods to parameterize the reverse process, we also consider general distributions $p_\theta(\sigma'_t|X_t)$ whose support are the whole $S_n$, *i.e.*, $\mathcal{T} = S_n$.

**The PL Distribution.** A popular distribution over $S_n$ is the Plackett-Luce (PL) distribution (Plackett, 1975; Luce, 1959), which is constructed from $n$ scores $\mathbf{s} = (s_1, \ldots, s_n)$ as follows,

$$p_{\text{PL}}(\sigma) = \prod_{i=1}^{n} \exp\left(s_{\sigma(i)}\right) / \left(\sum_{j=i}^{n} \exp\left(s_{\sigma(j)}\right)\right), \tag{8}$$

for all $\sigma \in S_n$. Intuitively, $(s_1, \ldots, s_n)$ represents the preference given to each index in $[n]$. To sample from $\text{PL}_{\mathbf{s}}$, we first sample $\sigma(1)$ from $\text{Cat}(n, \text{softmax}(\mathbf{s}))$. Then we remove $\sigma(1)$ from the list and sample $\sigma(2)$ from the categorical distribution corresponding to the rest of the scores (logits). We continue in this manner until we have sampled $\sigma(1), \ldots, \sigma(n)$. By Cao et al. (2007), the mode of the PL distribution is the permutation that sorts $\mathbf{s}$ in descending order. However, the PL distribution is not very expressive. In particular, we have the following result, and the proof is given in Appendix E.

**Proposition 1.** *The PL distribution cannot represent a delta distribution over $S_n$.*

**The Generalized PL (GPL) Distribution.** We then propose a generalization of the PL distribution, referred to as *Generalized Plackett-Luce (GPL) Distribution*. Unlike the PL distribution, which uses a set of $n$ scores, the GPL distribution uses $n^2$ scores $\{\mathbf{s}_1, \cdots, \mathbf{s}_n\}$, where each $\mathbf{s}_i = \{s_{i,1}, \ldots, s_{i,n}\}$ consists of $n$ scores. The GPL distribution is constructed as follows,

$$p_{\text{GPL}}(\sigma) := \prod_{i=1}^{n} \exp\left(s_{i,\sigma(i)}\right) / \left(\sum_{j=i}^{n} \exp\left(s_{i,\sigma(j)}\right)\right). \tag{9}$$

Sampling of the GPL distribution begins with sampling $\sigma(1)$ using $n$ scores $\mathbf{s}_1$. For $2 \leq i \leq n$, we remove $i-1$ scores from $\mathbf{s}_i$ that correspond to $\sigma(1), \ldots, \sigma(i-1)$ and sample $\sigma(i)$ from a categorical distribution constructed from the remaining $n - i + 1$ scores in $\mathbf{s}_i$. It is important to note that the family of PL distributions is a strict subset of the GPL family. Since the GPL distribution has more parameters than the PL distribution, it is expected to be more expressive. In fact, we prove the following significant result, and the proof is given in Appendix E.

**Theorem 2.** *The reverse diffusion process parameterized using the GPL distribution in Eq. (9) can model any distribution over $S_n$.*

### 3.3 NETWORK ARCHITECTURE AND TRAINING

We now briefly introduce how to use neural networks to parameterize the above distributions used in the reverse process. At any time $t$, given $X_t \in \mathbb{R}^{n \times d}$, we use a neural network with parameters $\theta$ to construct $p_\theta(\sigma'_t|X_t)$. In particular, we treat $n$ rows of $X_t$ as $n$ tokens and use a Transformer architecture along with the time embedding of $t$ and the positional encoding to predict the previously mentioned scores. For example, for the GPL distribution, to predict $n^2$ scores, we introduce $n$ dummy tokens that correspond to the $n$ permuted output positions. We then perform a few layers of masked self-attention ($2n \times 2n$) to obtain the token embedding $Z_1 \in \mathbb{R}^{n \times d_{\text{model}}}$ corresponding to $n$ input tokens and $Z_2 \in \mathbb{R}^{n \times d_{\text{model}}}$ corresponding to $n$ dummy tokens. Finally, the GPL score matrix is obtained as $S_\theta = Z_1 Z_2^\top \in \mathbb{R}^{n \times n}$. Since the aforementioned distributions have different numbers of scores, the specific architectures of the Transformer differ. We provide more details in Appendix B.

To learn the diffusion model, we maximize the following variational lower bound:

$$\mathbb{E}_{p_{\text{data}}(X_0, \mathcal{X})}\left[\log p_\theta(X_0|\mathcal{X})\right] \geq \mathbb{E}_{p_{\text{data}}(X_0, \mathcal{X})q(X_{1:T}|X_0, \mathcal{X})}\left[\log p(X_T|\mathcal{X}) + \sum_{t=1}^{T} \log \frac{p_\theta(X_{t-1}|X_t)}{q(X_t|X_{t-1})}\right]. \tag{10}$$

In practice, one can draw samples to obtain the Monte Carlo estimation of the lower bound. Due to the complexity of shuffling transition in the forward process, we can not obtain $q(X_t|X_0)$ analytically,

as is done in common diffusion models. Therefore, we have to run the forward process to collect samples. Fortunately, it is efficient as the forward process only involves shuffling integers. We include more training details in Appendix G.

Note that most existing diffusion models, such as those proposed by Ho et al. (2020) and Austin et al. (2023), use an equivalent form of the above variational bound, which involves the analytical KL divergence between the posterior $q(X_{t-1}|X_t, X_0)$ and $p_\theta(X_{t-1}|X_t)$ for variance control. However, this variational bound cannot be applied to $S_n$ because the transitions are not Gaussian, and $q(X_{t-1}|X_t, X_0)$ is generally unavailable for most shuffling methods. Most existing diffusion models also sample a random timestep of the loss. While this technique is also available in our framework, it introduces more variance and does not improve efficiency all the time. And for riffle shuffles, the trajectory is usually short enough that we can compute the loss on the whole trajectory. A detailed discussion can be found in Appendix C.

## 3.4 Denoising Schedule via Merging Reverse Steps

If one merges some steps in the reverse process, sampling and learning would be faster and more memory efficient. The variance of the training loss could also be reduced. Specifically, at time $t$ of the reverse process, instead of predicting $p_\theta(X_{t-1}|X_t)$, we can predict $p_\theta(X_{t'}|X_t)$ for any $0 \leq t' < t$. Given a sequence of timesteps $0 = t_0 < \cdots < t_k = T$, we can now model the reverse process as

$$p_\theta(X_{t_0}, \ldots, X_{t_k}|\mathcal{X}) = p(X_T|\mathcal{X}) \prod_{i=1}^k p_\theta(X_{t_{i-1}}|X_{t_i}). \tag{11}$$

To align with the literature of diffusion models, we call the list $[t_0, \ldots, t_k]$ the *denoising schedule*. After incorporating the denoising schedule in Eq. (10), we obtain the loss function:

$$\mathcal{L}(\theta) = \mathbb{E}_{p_{\text{data}}(X_0, \mathcal{X})} \mathbb{E}_{q(X_{1:T}|X_0, \mathcal{X})} \left[ -\log p(X_T|\mathcal{X}) - \sum_{i=1}^k \log \frac{p_\theta(X_{t_{i-1}}|X_{t_i})}{q(X_{t_i}|X_{t_{i-1}})} \right]. \tag{12}$$

Note that although we may not have the analytical form of $q(X_{t_i}|X_{t_{i-1}})$, we can draw samples from it. Merging is feasible if the support of $p_\theta(X_{t_{i-1}}|X_{t_i})$ is equal or larger than the support of $q(X_{t_i}|X_{t_{i-1}})$; otherwise, the inverse of some forward permutations would be almost surely unrecoverable. Therefore, we can implement a non-trivial denoising schedule (*i.e.*, $k < T$), when $p_\theta(\sigma_t'|X_t)$ follows the PL or GPL distribution, as they have whole $S_n$ as their support. However, merging is not possible for inverse shuffling methods, as their support is smaller than that of the corresponding multi-step forward shuffling. To design a successful denoising schedule, we first describe the intuitive principles and then provide some theoretical insights. 1) The length of forward diffusion $T$ should be minimal so long as the forward process approaches the uniform distribution. 2) If distributions of $X_t$ and $X_{t+1}$ are similar, we should merge these two steps. Otherwise, we should not merge them, as it would make the learning problem harder.

To quantify the similarity between distributions shown in 1) and 2), the TV distance is commonly used in the literature. In particular, we can measure $D_{\text{TV}}(q^{(t)}, q^{(t')})$ for $t \neq t'$ and $D_{\text{TV}}(q^{(t)}, u)$, where $q^{(t)}$ is the distribution at time $t$ in the forward process and $u$ is the uniform distribution. For riffle shuffles, the total variation distance can be computed exactly. Specifically, we first introduce the *Eulerian Numbers* $A_{n,r}$ (OEIS Foundation Inc., 2024), *i.e.*, the number of permutations in $S_n$ that have exactly $r$ rising sequences where $1 \leq r \leq n$. $A_{n,r}$ can be computed using the following recursive formula $A_{n,r} = rA_{n-1,r} + (n - r + 1)A_{n-1,r-1}$ where $A_{1,1} = 1$. We then have the following result. The proof is given in Appendix F.

**Proposition 3.** *Let $t \neq t'$ be positive integers. Then*

$$D_{\text{TV}}\left(q_{\text{RS}}^{(t)}, q_{\text{RS}}^{(t')}\right) = \frac{1}{2} \sum_{r=1}^n A_{n,r} \left| \frac{1}{2^{tn}} \binom{n + 2^t - r}{n} - \frac{1}{2^{t'n}} \binom{n + 2^{t'} - r}{n} \right|, \tag{13}$$

*and*

$$D_{\text{TV}}\left(q_{\text{RS}}^{(t)}, u\right) = \frac{1}{2} \sum_{r=1}^n A_{n,r} \left| \frac{1}{2^{tn}} \binom{n + 2^t - r}{n} - \frac{1}{n!} \right|. \tag{14}$$

Note that Eq. (14) was originally given by Kanungo (2020). We restate it here for completeness. Once the Eulerian numbers are precomputed, the TV distances can be computed in $O(n)$ time instead

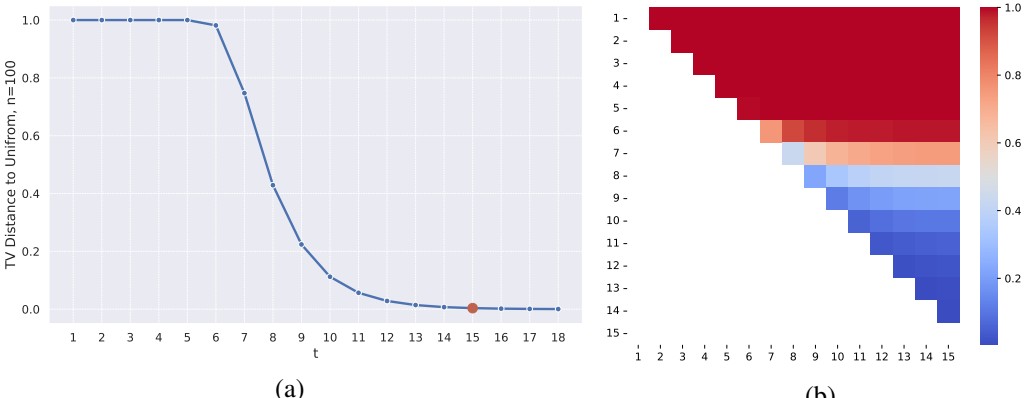

Figure 2: (a) $D_{\text{TV}}\big(q_{\text{RS}}^{(t)}, u\big)$ computed using Eq. (14). We choose $T = 15$ (red dot) based on the threshold 0.005. (b) A heatmap for $D_{\text{TV}}\big(q_{\text{RS}}^{(t)}, q_{\text{RS}}^{(t')}\big)$ for $n = 100$ and $1 \leq t < t' \leq 15$, computed using Eq. (13). Rows are $t$ and columns are $t'$. We choose the denoising schedule $[0, 8, 10, 15]$.

of $O(n!)$. Through extensive experiments, we have the following empirical observation. For the principle 1), choosing $T$ so that $D_{\text{TV}}\big(q_{\text{RS}}^{(T)}, u\big) \approx 0.005$ yields good results. For the principle 2), a denoising schedule $[t_0, \dots, t_k]$ with $D_{\text{TV}}\big(q_{\text{RS}}^{(t_i)}, q_{\text{RS}}^{(t_{i+1})}\big) \approx 0.3$ for most $i$ works well. We show an example on sorting $n = 100$ four-digit MNIST images in Fig. 2.

### 3.5 REVERSE PROCESS DECODING

We now discuss how to decode predictions from the reverse process at test time. In practice, one is often interested in the most probable state or a few states with high probabilities under $p_\theta(X_0|\mathcal{X})$. However, since we can only draw samples from $p_\theta(X_0|\mathcal{X})$ via running the reverse process, exact decoding is intractable. The simplest approximated method is greedy search, *i.e.*, successively finding the mode or an approximated mode of $p_\theta(X_{t_{i-1}}|X_{t_i})$. Another approach is beam search, which maintains a dynamic buffer of $k$ candidates with highest probabilities. Nevertheless, for one-step reverse transitions like the GPL distribution, even finding the mode is intractable. To address this, we employ a hierarchical beam search that performs an inner beam search within $n^2$ scores at each step of the outer beam search. Further details are provided in Appendix D.

## 4 EXPERIMENTS

We now demonstrate the general applicability and effectiveness of our model through solving a variety of tasks, including sorting 4-digit MNIST numbers, jigsaw puzzles, and traveling salesman problems (TSPs). Additional details, including an additional synthetic experiment that compares our method with other discrete diffusion models, are provided in Appendix G due to space constraints.

### 4.1 SORTING 4-DIGIT MNIST IMAGES

We first evaluate our SymmetricDiffusers on the four-digit MNIST sorting benchmark, a well-established testbed for differentiable sorting (Blondel et al., 2020; Cuturi et al., 2019; Grover et al., 2018; Kim et al., 2024; Petersen et al., 2021; 2022). Each four-digit image in this benchmark is obtained by concatenating 4 individual images from MNIST, and our task is to sort $n$ four-digit MNIST numbers. For evaluation, we employ several metrics to compare methods, including Kendall-Tau coefficient (measuring the correlation between rankings), accuracy (percentage of images perfectly reassembled), and correctness (percentage of pieces that are correctly placed).

**Ablation Study.** We conduct an ablation study to verify our design choices for reverse transition and decoding strategies. As shown in Table 3, when using riffle shuffles as the forward process, combining PL with either beam search (BS) or greedy search yields good results in terms of Kendall-Tau and correctness metrics. In contrast, the IRS (inverse riffle shuffle) method, along with greedy search, performs poorly across all metrics, showing the limitations of IRS in handling complicated sorting tasks. At the same time, combining GPL and BS achieves the best accuracy in correctly sorting the entire sequence of images. Finally, we see that random transpositions (RT) and random insertions

Table 1: Results (averaged over 5 runs) on solving the jigsaw puzzle on Noisy MNIST and CIFAR10.

| Method | Metrics | Noisy MNIST | | | | | CIFAR-10 | | |
|---|---|---|---|---|---|---|---|---|---|
| | | $2 \times 2$ | $3 \times 3$ | $4 \times 4$ | $5 \times 5$ | $6 \times 6$ | $2 \times 2$ | $3 \times 3$ | $4 \times 4$ |
| Gumbel-Sinkhorn Network (Mena et al., 2018) | Kendall-Tau ↑ | 0.9984 | 0.6908 | 0.3578 | 0.2430 | **0.1755** | 0.8378 | 0.5044 | **0.4016** |
| | Accuracy (%) | 99.81 | 44.65 | 00.86 | 0.00 | 0.00 | 76.54 | 6.07 | 0.21 |
| | Correct (%) | 99.91 | 80.20 | 49.51 | 26.94 | 14.91 | 86.10 | 43.59 | 25.31 |
| | RMSE ↓ | **0.0022** | 0.1704 | 0.4572 | 0.8915 | 1.0570 | 0.3749 | 0.9590 | 1.0960 |
| | MAE ↓ | 0.0003 | 0.0233 | 0.1005 | 0.3239 | 0.4515 | 0.1368 | 0.5320 | 0.6873 |
| DiffSort (Petersen et al., 2022) | Kendall-Tau ↑ | 0.9931 | 0.3054 | 0.0374 | 0.0176 | 0.0095 | 0.6463 | 0.1460 | 0.0490 |
| | Accuracy (%) | 99.02 | 5.56 | 0.00 | 0.00 | 0.00 | 59.18 | 0.96 | 0.00 |
| | Correct (%) | 99.50 | 42.25 | 10.77 | 6.39 | 3.77 | 75.48 | 27.87 | 12.27 |
| | RMSE ↓ | 0.0689 | 1.0746 | 1.3290 | 1.4883 | 1.5478 | 0.7389 | 1.2691 | 1.3876 |
| | MAE ↓ | 0.0030 | 0.4283 | 0.6531 | 0.8204 | 0.8899 | 0.2800 | 0.8123 | 0.9737 |
| Error-free DiffSort (Kim et al., 2024) | Kendall-Tau ↑ | 0.9899 | 0.2014 | 0.0100 | 0.0034 | -0.0021 | 0.6604 | 0.1362 | 0.0318 |
| | Accuracy (%) | 98.62 | 0.82 | 0.00 | 0.00 | 0.00 | 60.96 | 0.68 | 0.00 |
| | Correct (%) | 99.28 | 32.65 | 7.40 | 4.39 | 2.50 | 75.99 | 26.75 | 10.33 |
| | RMSE ↓ | 0.0814 | 1.1764 | 1.3579 | 1.5084 | 1.5606 | 0.7295 | 1.2820 | 1.4095 |
| | MAE ↓ | 0.0041 | 0.5124 | 0.6818 | 0.8424 | 0.9041 | 0.2731 | 0.8260 | 0.9990 |
| Symmetric Diffusers (Ours) | Kendall-Tau ↑ | **0.9992** | **0.8126** | **0.4859** | **0.2853** | 0.1208 | **0.9023** | **0.8363** | 0.2518 |
| | Accuracy (%) | **99.88** | **57.38** | **1.38** | 0.00 | 0.00 | **90.15** | **70.94** | **0.64** |
| | Correct (%) | **99.94** | **86.16** | **58.51** | **37.91** | **18.54** | **92.99** | **86.84** | **34.69** |
| | RMSE ↓ | 0.0026 | **0.0241** | **0.1002** | **0.2926** | **0.4350** | **0.3248** | **0.3892** | **0.8953** |
| | MAE ↓ | **0.0001** | **0.0022** | **0.0130** | **0.0749** | **0.1587** | **0.0651** | **0.0977** | **0.5044** |

Table 2: Results (averaged over 5 runs) on the four-digit MNIST sorting benchmark. For $n = 200$, due to efficiency reasons, we use PL for the reverse process, and we randomly sample a timestep when computing the loss (see Appendix C.2).

| Method | Metrics | Sequence Length | | | | | | | | |
|---|---|---|---|---|---|---|---|---|---|---|
| | | 3 | 5 | 7 | 9 | 15 | 32 | 52 | 100 | 200 |
| DiffSort (Petersen et al., 2022) | Kendall-Tau ↑ | 0.930 | 0.898 | 0.864 | 0.801 | 0.638 | 0.535 | 0.341 | 0.166 | 0.107 |
| | Accuracy (%) | 93.8 | 83.9 | 71.5 | 52.2 | 10.3 | 0.2 | 0.0 | 0.0 | 0.0 |
| | Correct (%) | 95.8 | 92.9 | 90.1 | 85.2 | 82.3 | 61.8 | 42.8 | 23.2 | 15.3 |
| Error-free DiffSort (Kim et al., 2024) | Kendall-Tau ↑ | 0.974 | **0.967** | **0.962** | **0.952** | **0.938** | **0.879** | 0.170 | 0.140 | 0.002 |
| | Accuracy (%) | 97.7 | 95.3 | 92.9 | 89.6 | **83.1** | **57.1** | 0.0 | 0.0 | 0.0 |
| | Correct (%) | 98.4 | **97.7** | **97.2** | **96.3** | **95.1** | **90.1** | 24.2 | 20.1 | 0.8 |
| Symmetric Diffusers (Ours) | Kendall-Tau ↑ | **0.976** | **0.967** | 0.959 | 0.950 | 0.932 | 0.858 | **0.786** | **0.641** | **0.453** |
| | Accuracy (%) | **98.0** | **95.5** | **92.9** | **90.0** | 82.6 | 55.1 | **27.4** | **4.5** | **0.1** |
| | Correct (%) | **98.5** | 97.6 | 96.8 | 96.1 | 94.5 | 88.3 | **82.1** | **69.3** | **52.2** |

(RI) are both out of memory for large instances due to their long mixing time. Given that accuracy is the most challenging metric to improve, we select riffle shuffles, GPL and BS for all remaining experiments, unless otherwise specified. More ablation study (*e.g.*, denoising schedule) is provided in Appendix G.3.

**Full Results.** From Table 2, we can see that Error-free DiffSort achieves the best performance in sorting sequences with lengths up to 32. However, its performance declines considerably with longer sequences (e.g., those exceeding 52 in length). Meanwhile, DiffSort performs the worst due to the error accumulation of its soft differentiable swap function (Kim et al., 2024; Petersen et al., 2021). In contrast, our method is on par with Error-free DiffSort in sorting short sequences and significantly outperforms others on long sequences.

## 4.2 JIGSAW PUZZLE

We then explore image reassembly from segmented "jigsaw" puzzles (Mena et al., 2018; Noroozi & Favaro, 2016; Santa Cruz et al., 2017). We evaluate the performance using the MNIST and the CIFAR10 datasets, which comprises puzzles of up to $6 \times 6$ and $4 \times 4$ pieces respectively. We add slight noise to pieces from the MNIST dataset to ensure background pieces are distinctive. To evaluate our models, we use Kendall-Tau coefficient, accuracy, correctness, RMSE (root mean square error of reassembled images), and MAE (mean absolute error) as metrics.

Table 1 presents results comparing our method with the Gumbel-Sinkhorn Network (Mena et al., 2018), Diffsort (Petersen et al., 2022), and Error-free Diffsort (Kim et al., 2024). DiffSort and Error-free DiffSort are primarily designed for sorting high-dimensional ordinal data which have clearly different patterns. Since jigsaw puzzles on MNIST and CIFAR10 contain pieces that are visually similar, these methods do not perform well. The Gumbel-Sinkhorn performs better for tasks

Table 3: Ablation study on transitions of reverse diffusion and decoding strategies. Results are averaged over three runs on sorting 52 four-digit MNIST images. GPL: generalized Plackett-Luce; IRS: inverse riffle shuffle; RT: random transposition; IT: inverse transposition; RI: random insertion; II: inverse insertion.

| Forward | Riffle Shuffles | | | | | RT | RI |
|---|---|---|---|---|---|---|---|
| Reverse | GPL + BS | GPL + Greedy | PL + Greedy | PL + BS | IRS + Greedy | IT + Greedy | II + Greedy |
| Kendall-Tau ↑ | 0.786 | **0.799** | **0.799** | 0.797 | 0.390 | | |
| Accuracy (%) | **27.4** | 24.4 | 26.4 | 26.4 | 0.6 | Out of Memory | |
| Correct (%) | 82.1 | 81.6 | **83.3** | 83.1 | 44.6 | | |

Table 4: Results on TSP-20 and TSP-50. We compare our method with OR solvers such as Concorde (Applegate et al., 2006), LKH-3 (Helsgaun, 2017), and 2-Opt (Lin & Kernighan, 1973), as well as learning-based approaches including GCN (Joshi et al., 2019) and DIFUSCO (Sun & Yang, 2023) on 20-node and 50-node TSP instances. An asterisk (*) indicates that post-processing heuristics were removed to ensure a fair comparison. Baselines for TSP-50 are taken from Sun & Yang (2023).

| Method | | TSP-20 | | TSP-50 | |
|---|---|---|---|---|---|
| | | Tour Length ↓ | Optimality Gap (%) ↓ | Tour Length ↓ | Optimality Gap (%) ↓ |
| **OR Solvers** | Concorde | **3.84** | **0.00** | **5.69** | **0.00** |
| | LKH-3 | **3.84** | **0.00** | **5.69** | **0.00** |
| | 2-Opt | 4.02 | 4.64 | 5.86 | 2.95 |
| **Learning-** | GCN | **3.85*** | 0.21* | 5.87 | 3.10 |
| **Based** | DIFUSCO | 3.88* | 1.07* | **5.70** | **0.10** |
| **Models** | **Ours** | **3.85** | 0.18 | 5.71 | 0.41 |

involving fewer than $4 \times 4$ pieces. In more challenging scenarios (*e.g.*, $5 \times 5$ and $6 \times 6$), our method significantly outperforms all competitors.

## 4.3 THE TRAVELLING SALESMAN PROBLEM

At last, we explore the travelling salesman problem (TSP) to demonstrate the general applicability of our model. TSPs are classical NP-complete combinatorial optimization problems which are solved using integer programming or heuristic solvers (Arora, 1998; Gonzalez, 2007). There exists a vast literature on learning-based models to solve TSPs (Kipf & Welling, 2017; Kool et al., 2019; Joshi et al., 2019; 2021; Bresson & Laurent, 2021; Kwon et al., 2021; Fu et al., 2021; Qiu et al., 2022; Kim et al., 2023; Sun & Yang, 2023; Min et al., 2024; Sanokowski et al., 2024). They often focus on the Euclidean TSPs, which are formulated as follows. Let $V = \{v_1, \ldots, v_n\}$ be points in $\mathbb{R}^2$. We need to find some $\sigma \in S_n$ such that $\sum_{i=1}^{n} \|v_{\sigma(i)} - v_{\sigma(i+1)}\|_2$ is minimized, where we let $\sigma(n+1) := \sigma(1)$.

We compare with operations research (OR) solvers and other learning based approaches on TSP instances with 20 nodes and 50 nodes. The metrics are the total tour length and the optimality gap. Given the ground truth (GT) length produced by the best OR solver, the optimality gap is given by $\big(\text{predicted length} - (\text{GT length})\big)/(\text{GT length})$. As shown in Table 4, SymmetricDiffusers achieves comparable results with both OR solvers and the state-of-the-art learning-based methods. Further experiment details are provided in Appendix G.

## 5 CONCLUSION

In this paper, we introduce a novel discrete diffusion model over finite symmetric groups. We identify the riffle shuffle as an effective forward transition and provide empirical rules for selecting the diffusion length. Additionally, we propose a generalized PL distribution for the reverse transition, which is provably more expressive than the PL distribution. We further introduce a theoretically grounded "denoising schedule" to improve sampling and learning efficiency. Extensive experiments verify the effectiveness of our proposed model. Despite significantly surpassing the performance of existing methods on large instances, our method still has limitations in larger scales. In the future, we would like to explore methods to improve scalability even more. We would also like to explore how we can fit other modern techniques in diffusion models like concrete scores (Meng et al., 2023) and score entropy (Lou et al., 2024) into our shuffling dynamics. Finally, we are interested in generalizing our model to general finite groups and exploring diffusion models on Lie groups.

ACKNOWLEDGMENTS

We thank the anonymous reviewers for their helpful comments. This work was funded, in part, by the NSERC DG Grant (No. RGPIN-2022-04636), the Vector Institute for AI, and Canada CIFAR AI Chair. Resources used in preparing this research were provided, in part, by the Province of Ontario, the Government of Canada through the Digital Research Alliance of Canada `alliance.can.ca`, and companies sponsoring the Vector Institute `www.vectorinstitute.ai/#partners`, and Advanced Research Computing at the University of British Columbia. Additional hardware support was provided by John R. Evans Leaders Fund CFI grant.

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

## A    ADDITIONAL DETAILS OF THE GSR RIFFLE SHUFFLE MODEL

There are many equivalent definitions of the GSR riffle shuffle. Here we also introduce the *Geometric Description* (Bayer & Diaconis, 1992), which is easy to implement (and is how we implement riffle shuffles in our experiments). We first sample $n$ points in the unit interval $[0, 1]$ uniformly and independently, and suppose the points are labeled in order as $x_1 < x_2 < \cdots < x_n$. Then, the permutation that sorts the points $\{2x_1\}, \ldots, \{2x_n\}$ follows the GSR distribution, where $\{x\} := x - \lfloor x \rfloor$ is the fractional part of $x$.

## B    DETAILS OF OUR NETWORK ARCHITECTURE

We now discuss how to use neural networks to produce the parameters of the distributions discussed in Section 3.2.1 and 3.2.2. Fix time $t$, and suppose $X_t = \left(\mathbf{x}_1^{(t)}, \ldots, \mathbf{x}_n^{(t)}\right)^\top \in \mathbb{R}^{n \times d}$. Let $\texttt{encoder}_\theta$ be an object-specific encoder such that $\texttt{encoder}_\theta(X_t) \in \mathbb{R}^{n \times d_{\text{model}}}$. For example, $\texttt{encoder}_\theta$ can be a CNN if $X_t$ is an image. Let

$$Y_t := \texttt{encoder}_\theta(X_t) + \texttt{time\_embd}(t) = \left(\mathbf{y}_1^{(t)}, \ldots, \mathbf{y}_n^{(t)}\right)^\top \in \mathbb{R}^{n \times d_{\text{model}}}, \tag{15}$$

where $\texttt{time\_embd}$ is the sinusoidal time embedding. Then, we would like to feed the embeddings into a Transformer encoder (Vaswani et al., 2023). Let $\texttt{transformer\_encoder}_\theta$ be the encoder part of the Transformer architecture. However, each of the distributions we discussed previously has different number of parameters, so we will have to discuss them separately.

**Inverse Transposition.**    For Inverse Transposition, we have $n + 1$ parameters. To obtain $n + 1$ tokens from $\texttt{transformer\_encoder}_\theta$, we append a dummy token of 0's to $Y_t$. Then we input $\left(\mathbf{y}_1^{(t)}, \ldots, \mathbf{y}_n^{(t)}, 0\right)^\top$ into $\texttt{transformer\_encoder}_\theta$ to obtain $Z \in \mathbb{R}^{(n+1) \times d_{\text{model}}}$. Finally, we apply an MLP to obtain $(s_1, \ldots, s_n, k) \in \mathbb{R}^{n+1}$.

**Inverse Insertion, Inverse Riffle Shuffle, PL Distribution.**    These three distributions all require exactly $n$ parameters, so we can directly feed $Y_t$ into $\texttt{transformer\_encoder}_\theta$. Let the output of $\texttt{transformer\_encoder}_\theta$ be $Z \in \mathbb{R}^{n \times d_{\text{model}}}$, where we then apply an MLP to obtain the scores $\mathbf{s}_\theta \in \mathbb{R}^n$.

**The GPL Distribution.**    The GPL distribution requires $n^2$ parameters. We first append $n$ dummy tokens of 0's to $Y_t$, with the intent that the $j^{\text{th}}$ dummy token would learn information about the $j^{\text{th}}$ column of the GPL parameter matrix, which represents where the $j^{\text{th}}$ component should be placed. We then pass $\left(\mathbf{y}_1^{(t)}, \ldots, \mathbf{y}_n^{(t)}, 0, \ldots, 0\right)^\top \in \mathbb{R}^{2n \times d_{\text{model}}}$ to $\texttt{transformer\_encoder}_\theta$. When computing attention, we further apply a $2n \times 2n$ attention mask

$$M := \begin{bmatrix} 0 & A \\ 0 & B \end{bmatrix}, \text{ where } A \text{ is an } n \times n \text{ matrix of } -\infty, \ B = \begin{bmatrix} -\infty & -\infty & \cdots & -\infty \\ 0 & -\infty & \cdots & -\infty \\ \vdots & \vdots & \ddots & \vdots \\ 0 & 0 & \cdots & -\infty \end{bmatrix} \text{ is } n \times n.$$

The reason for having $B$ as an upper triangular matrix of $-\infty$ is that information about the $j^{\text{th}}$ component should only require information from the previous components. Let

$$\texttt{transformer\_encoder}_\theta(Y_t, M) = \begin{bmatrix} Z_1 \\ Z_2 \end{bmatrix},$$

where $Z_1, Z_2 \in \mathbb{R}^{n \times d_{\text{model}}}$. Finally, we obtain the GPL parameter matrix as $S_\theta = Z_1 Z_2^\top \in \mathbb{R}^{n \times n}$.

For hyperparameters, we refer the readers to Appendix G.5.

## C  DISCUSSIONS ON OTHER FORMS OF THE LOSS

### C.1  USING KL DIVERGENCE

Many diffusion models will rewrite the variational bound Eq.(10) in the following equivalent form of KL divergences to reduce the variance (Austin et al., 2023; Ho et al., 2020):

$$\mathbb{E}_{p_{\text{data}}(X_0,\mathcal{X})q(X_{1:T}|X_0)}\Bigg[D_{\text{KL}}(q(X_t|X_0) \parallel p(X_T|\mathcal{X}))$$

$$+ \sum_{t>1} D_{\text{KL}}(q(X_{t-1}|X_t,X_0) \parallel p_\theta(X_{t-1}|X_t)) - \log p_\theta(X_0|X_1)\Bigg] \quad (16)$$

However, we cannot use this objective for $S_n$ in most cases. In particular, since

$$q(X_{t-1}|X_t,X_0) = \frac{q(X_t|X_{t-1})q(X_{t-1}|X_0)}{q(X_t|X_0)}, \quad (17)$$

we can only derive the analytical form of $q(X_{t-1}|X_t,X_0)$ if we know the form of $q(X_t|X_0)$. However, $q(X_t|X_0)$ is unavailable for most shuffling methods used in the forward process except for the riffle shuffles. For riffle shuffle, $q(X_t|X_0)$ is actually available and permits efficient sampling (Bayer & Diaconis, 1992). However, $D_{\text{KL}}(q(X_{t-1}|X_t,X_0) \parallel p_\theta(X_{t-1}|X_t))$ still does not have an analytical form, unlike in common diffusion models. As a result, we cannot use mean/score parameterization (Ho et al., 2020; Song et al., 2021) commonly employed in the continuous setting. Therefore, we need to rewrite the KL term as follows and resort to Monte Carlo (MC) estimation,

$$\mathbb{E}_{q(X_t|X_0)}\Big[D_{\text{KL}}(q(X_{t-1}|X_t,X_0) \parallel p_\theta(X_{t-1}|X_t))\Big]$$

$$= \mathbb{E}_{q(X_t|X_0)}\Bigg[\sum_{X_{t-1}} \frac{q(X_t|X_{t-1})q(X_{t-1}|X_0)}{q(X_t|X_0)} \cdot \log \frac{q(X_{t-1}|X_t,X_0)}{p_\theta(X_{t-1}|X_t)}\Bigg]$$

$$= \mathbb{E}_{q(X_t|X_0)}\Bigg[\sum_{X_{t-1}} q(X_{t-1}|X_0) \cdot \frac{q(X_t|X_{t-1})}{q(X_t|X_0)} \cdot \log \frac{q(X_{t-1}|X_t,X_0)}{p_\theta(X_{t-1}|X_t)}\Bigg]$$

$$= \mathbb{E}_{q(X_t|X_0)}\mathbb{E}_{q(X_{t-1}|X_0)}\Bigg[\frac{q(X_t|X_{t-1})}{q(X_t|X_0)} \cdot \log \frac{q(X_{t-1}|X_t,X_0)}{p_\theta(X_{t-1}|X_t)}\Bigg]. \quad (18)$$

Note that $X_t \sim q(X_t|X_0)$ and $X_{t-1} \sim q(X_{t-1}|X_0)$ are drawn *independently*. However, there is a high chance that $q(X_t|X_{t-1}) = 0$ for the $X_t$ and $X_{t-1}$ that are sampled. Consequently, if we only draw a few MC samples, the resulting estimator will likely be zero with zero-valued gradients, impeding the optimization of the training objective. Therefore, writing the loss in the form of KL divergences does not help in the case of discrete diffusion on $S_n$.

### C.2  SAMPLING A RANDOM TIMESTEP

Another technique that many diffusion models use is to randomly sample a timestep $t$ and just compute the loss at time $t$. Our framework also allows for randomly sampling one timestep and compute the loss as

$$\mathbb{E}_{p_{\text{data}}(X_0,\mathcal{X})}\mathbb{E}_t\mathbb{E}_{q(X_{t-1}|X_0)}\mathbb{E}_{q(X_t|X_{t-1})}\Big[-\log p_\theta(X_{t-1}|X_t)\Big], \quad (19)$$

omitting constant terms with respect to $\theta$. With a denoising schedule of $[t_0, \ldots, t_k]$, the loss is

$$\mathbb{E}_{p_{\text{data}}(X_0,\mathcal{X})}\mathbb{E}_i\mathbb{E}_{q(X_{t_{i-1}}|X_0)}\mathbb{E}_{q(X_{t_i}|X_{t_{i-1}})}\Big[-\log p_\theta(X_{t_{i-1}}|X_{t_i})\Big], \quad (20)$$

again omitting constant terms with respect to $\theta$. It is also worth noting that computing the loss on a subset of the trajectory could potentially introduce more variance during training, which leads to a tradeoff. For riffle shuffles, although we can sample $X_{t-1}$ directly for arbitrary timestep $t-1$ from

$X_0$ as previously mentioned in this section, the whole trajectory would be really short. For other shuffling methods, we would still have to run the entire forward process to sample from $q(X_{t-1}|X_0)$, which does not solve the inefficiency problem of other shuffling methods. Therefore, we opt to use the loss in Eq.(10) in most cases, and we would resort to Eq.(20) for riffle shuffling really large instances.

## D    ADDITIONAL DETAILS OF DECODING

**Greedy Search.**    At each timestep $t_i$ in the denoising schedule, we can greedily obtain or approximate the mode of $p_\theta(X_{t_{i-1}}|X_{t_i})$. We can then use the (approximated) mode $X_{t_{i-1}}$ for the next timestep $p_\theta(X_{t_{i-2}}|X_{t_{i-1}})$. Note that the final $X_0$ obtained using such a greedy heuristic may not necessarily be the mode of $p_\theta(X_0|\mathcal{X})$.

**Beam Search.**    We can use beam search to improve the greedy approach. The basic idea is that, at each timestep $t_i$ in the denoising schedule, we compute or approximate the top-$k$-most-probable results from $p_\theta(X_{t_{i-1}}|X_{t_i})$. For each of the top-$k$ results, we sample top-$k$ from $p_\theta(X_{t_{i-2}}|X_{t_{i-1}})$. Now we have $k^2$ candidates for $X_{t_{i-2}}$, and we only keep the top $k$ of the $k^2$ candidates.

However, it is not easy to obtain the top-$k$-most-probable results for some of the distributions. Here we provide an algorithm to approximate top-$k$ of the PL and the GPL distribution. Since the PL distribution is a strict subset of the GPL distribution, it suffices to only consider the GPL distribution with parameter matrix $S$. The algorithm for approximating top-$k$ of the GPL distribution is another beam search. We first pick the $k$ largest elements from the first row of $S$. For each of the $k$ largest elements, we pick $k$ largest elements from the second row of $S$, excluding the corresponding element picked in the first row. We now have $k^2$ candidates for the first two elements of a permutation, and we only keep the top-$k$-most-probable candidates. We then continue in this manner.

## E    THE EXPRESSIVENESS OF PL AND GPL

In this section, we prove the expressiveness results of the PL and GPL distribution. We first show that the PL distribution has limited expressiveness.

**Proposition 1.** *The PL distribution cannot represent a delta distribution over $S_n$.*

*Proof.* Assume for a contradiction that there exists some $\sigma \in S_n$ and $\mathbf{s}$ such that $\text{PL}_\mathbf{s} = \delta_\sigma$. Then we have

$$\prod_{i=1}^{n} \frac{\exp\left(s_{\sigma(i)}\right)}{\sum_{j=i}^{n} \exp\left(s_{\sigma(j)}\right)} = 1.$$

Since each of the term in the product is less than or equal to 1, we must have

$$\frac{\exp\left(s_{\sigma(i)}\right)}{\sum_{j=i}^{n} \exp\left(s_{\sigma(j)}\right)} = 1 \tag{21}$$

for all $i \in [n]$. In particular, we have

$$\frac{\exp\left(s_{\sigma(1)}\right)}{\sum_{j=1}^{n} \exp\left(s_{\sigma(j)}\right)} = 1,$$

which happens if and only if $s_{\sigma(j)} = -\infty$ for all $j \geq 2$. But this contradicts (21).    $\square$

We then prove that the reverse process using the GPL distribution can model any target distribution. We first introduce two lemmas.

**Lemma 4.** *The GPL distribution can represent any delta distribution on $S_n$ if allowing $-\infty$ scores.*

*Proof.* Fix $\sigma \in S_n$. For all $i \in [n]$, we let $s_{i,\sigma(i)} = 0$ and $s_{i,j} = -\infty$ for all $j \neq \sigma(i)$. Then it is clear that $\text{GPL}_{(s_{ij})} = \delta_\sigma$.    $\square$

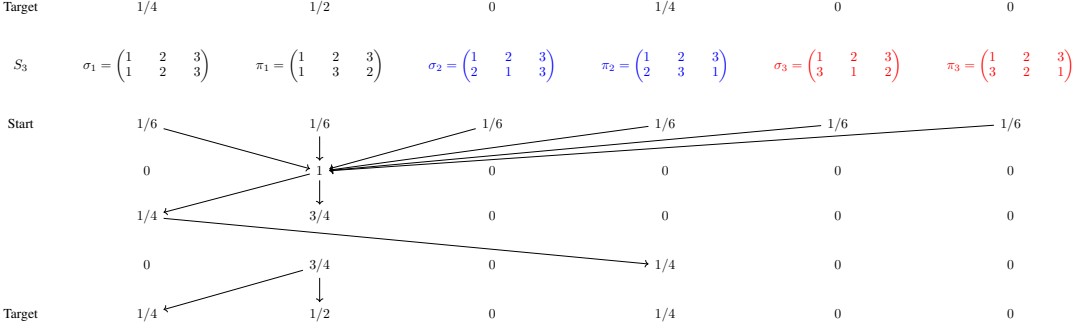

Figure 3: A simple example for the GPL expressiveness theorem on $S_3$.

**Lemma 5.** *Let $\sigma \in S_n$ and let $\pi := (n-1 \quad n) \circ \sigma$, where $(n-1 \quad n)$ is a transposition that swaps the last two indices and $\circ$ is function composition. That is, $\pi$ is obtained from $\sigma$ by swapping the last two components. Let $p$ be any probability distribution on $S_n$ whose support is a subset of $\{\sigma, \pi\}$. Then there exists scores $(s_{ij})_{i,j \in [n]}$ (possibly $-\infty$) such that $\mathrm{GPL}_{(s_{ij})} = p$.*

*Proof.* Note that we have

$$\pi = \begin{pmatrix} 1 & 2 & \cdots & n-1 & n \\ \sigma(1) & \sigma(2) & \cdots & \sigma(n) & \sigma(n-1) \end{pmatrix}.$$

For all $1 \leq i \leq n-2$, we let $s_{i,\sigma(i)} = 0$ and $s_{i,j} = -\infty$ for all $j \neq \sigma(i)$. Let $s_{n-1,\sigma(n-1)} = \ln p(\sigma)$ and let $s_{n-1,\sigma(n)} = \ln p(\pi)$. Finally, let $s_{n,j} = 0$ for all $j \in [n]$. It is then easy to verify that $\mathrm{GPL}_{(s_{ij})} = p$. $\qquad \square$

We then state the main expressiveness theorem.

**Theorem 6.** *Let $Y_0 \in S_n$ be a random variable with arbitrary distribution $q(Y_0)$. Let $p$ be any distribution over $S_n$. Then there exists some $k \in \mathbb{N}$ and random variables $Y_1, \ldots, Y_k$ with GPL (allowing $-\infty$ scores) transition distributions $q(Y_i \mid Y_{i-1})$ for each $i \in [k]$ such that $q(Y_k) = p$.*

Before proceeding to the proof, we first provide a small example illustrating the construction we are going to use. Consider Fig. 3 on $S_3$. Suppose we start with $q(Y_0)$ being the uniform distribution, and the target distribution $p$ is listed at the top row of the diagram. We partition $S_3$ into 3 pairs $(\sigma_i, \pi_i)$ indicated by their color in the diagram. The permutations within each pair differ by one swap of the last two indices, so each pair is the pair considered in Lemma 5. The first step is to concentrate all probability mass on $\pi_1$ using GPL and Lemma 4. Now note that we only need $1/4 + 1/2 = 3/4$ probability for the first pair $(\sigma_1, \pi_1)$, so there is a $1/4$ excess. We then use Lemma 5 to move the excess amount to $\sigma_1$. Then we use Lemma 4 to move the excess amount out of pair 1 to $\pi_2$ in pair 2. Finally, we use Lemma 5 to distribute the correct mass to $\sigma_1$ and $\pi_1$ from the $3/4$ that $\pi_1$ currently holds. We now present the formal construction.

*Proof.* For $\sigma, \pi \in S_n$, we say that $\sigma$ and $\pi$ are a pair if $\pi = (n-1 \quad n) \circ \sigma$. Note that we can partition $S_n$ into $n!/2$ disjoint pairs. We write the pairs as $(\sigma_1, \pi_1), \ldots, (\sigma_{n!/2}, \pi_{n!/2})$.

We now give an algorithm that explicitly constructs the transitions from $Y_0$ to the target distribution $p$. The intuition of the algorithm is that we distribute the probability mass for one pair of permutations at a time. The variable $p_{\text{leftover}}$ records how much mass we have yet to distribute.

1. Let $p_{\text{leftover}} := 1$. Let $q(Y_1 \mid Y_0) = \delta_{\pi_1}$.

2. Iterate through all pairs $(\sigma_i, \pi_i)$ for $i$ from 1 to $n!/2$:

   (a) Let $p_{\text{excess}} := p_{\text{leftover}} - p(\sigma_i) - p(\pi_i)$.
   
   (b) Define $q(Y_{3i-1} \mid Y_{3i-2})$ such that
   
   $$q(Y_{3i-1} = \sigma_i \mid Y_{3i-2} = \pi_i) = \frac{p_{\text{excess}}}{p_{\text{leftover}}},$$
   
   $$q(Y_{3i-1} = \pi_i \mid Y_{3i-2} = \pi_i) = 1 - \frac{p_{\text{excess}}}{p_{\text{leftover}}},$$
   
   and $q(Y_{3i-1} = \tau \mid Y_{3i-2} = \pi_i) = 0$ for all other $\tau \notin \{\sigma_i, \pi_i\}$. Let $q(Y_{3i-1} \mid Y_{3i-2} = \tau) = \delta_\tau$ for $\tau \notin \{\pi_i\}$.
   
   (c) Define $q(Y_{3i} \mid Y_{3i-1})$ such that $q(Y_{3i} \mid Y_{3i-1} = \sigma_i) = \delta_{\pi_{i+1}}$ and $q(Y_{3i} \mid Y_{3i-1} = \tau) = \delta_\tau$ for all $\tau \neq \sigma_i$.
   
   (d) Finally, define $q(Y_{3i+1} \mid Y_{3i})$ such that
   
   $$q(Y_{3i+1} = \sigma_i \mid Y_{3i} = \pi_i) = \frac{p(\sigma_i)}{1 - p_{\text{excess}}},$$
   
   $$q(Y_{3i+1} = \pi_i \mid Y_{3i} = \pi_i) = \frac{p(\pi_i)}{1 - p_{\text{excess}}},$$
   
   and $q(Y_{3i+1} = \tau \mid Y_{3i} = \pi_i) = 0$ for all other $\tau \notin \{\sigma_i, \pi_i\}$. Let $q(Y_{3i+1} \mid Y_{3i} = \tau) = \delta_\tau$ for $\tau \notin \{\pi_i\}$.
   
   (e) Update $p_{\text{leftover}} := p_{\text{excess}}$.

3. Return $q(Y_1 \mid Y_0), \ldots, q(Y_{(3n!/2)+1} \mid Y_{3n!/2})$.

Note that $q(Y_1 = \pi_1) = 1$. Also note that by Lemma 4 and 5, all transition distributions can be modeled by the GPL distribution. We claim that at the end of iteration $i$, we must have

(1) $p_{\text{leftover}} = \sum_{j=i+1}^{n!/2} p(\sigma_j) + p(\pi_j)$;

(2) $q(Y_{3i+1} = \sigma_i) = p(\sigma_i)$ and $q(Y_{3i+1} = \pi_i) = p(\pi_i)$;

(3) $q(Y_{3i+1} = \pi_{i+1}) = p_{\text{leftover}}$.

We proceed by induction on $i$. For $i = 1$, it is clear that $p_{\text{leftover}} = \sum_{j=2}^{n!/2} p(\sigma_j) + p(\pi_j)$ at the end of the first iteration. We also note that $q(Y_2 = \sigma_1) = p_{\text{excess}}$ and $q(Y_2 = \pi_1) = 1 - p_{\text{excess}} = p(\sigma_1) + p(\pi_1)$. After step 2(c) of the algorithm, we have $q(Y_3 = \sigma_1) = 0$, $q(Y_3 = \pi_1) = p(\sigma_1) + p(\pi_1)$, and $q(Y_3 = \pi_2) = p_{\text{excess}}$. Finally, after step 2(d) and 2(e), we get $q(Y_4 = \sigma_1) = p(\sigma_1)$, $q(Y_4 = \pi_1) = p(\pi_1)$, and $q(Y_4 = \pi_2) = p_{\text{leftover}}$.

For the inductive step, let $i \geq 2$. We know by the inductive hypothesis that at the start of iteration $i$, we have $p_{\text{leftover}} = \sum_{j=i}^{n!/2} p(\sigma_j) + p(\pi_j)$. So $0 \leq p_{\text{excess}} \leq 1$, and all transition distributions in iteration $i$ are well-defined. It is easy to verify that:

- After step 2(b), $q(Y_{3i-1} = \sigma_i) = p_{\text{excess}}$ and $q(Y_{3i-1} = \pi_i) = p(\sigma_i) + p(\pi_i)$.

- After step 2(c), $q(Y_{3i} = \sigma_i) = 0$, $q(Y_{3i} = \pi_i) = p(\sigma_i) + p(\pi_i)$, and $q(Y_{3i} = \pi_{i+1}) = p_{\text{excess}}$.

- After step 2(d)(e), $q(Y_{3i+1} = \sigma_i) = p(\sigma_i)$, $q(Y_{3i+1} = \pi_i) = p(\pi_i)$, and $q(Y_{3i+1} = \pi_{i+1}) = p_{\text{leftover}} = \sum_{j=i+1}^{n!/2} p(\sigma_j) + p(\pi_j)$.

The $p_{\text{leftover}}$ at the end of iteration $n!/2$ is $p_{\text{excess}} - p(\sigma_{n!/2}) - p(\pi_{n!/2}) = 0$. This finishes the induction. Finally, we observe that after iteration $i$, $q(Y_j = \sigma_i)$ and $q(Y_j = \pi_i)$ will never be changed for $j \geq i$. This finishes the proof. $\qquad\square$

Finally, Theorem 2, which is stated in the main paper, follows immediately from Theorem 6 by setting $q(Y_0)$ to be the uniform distribution and $p$ to be the target distribution over $S_n$.

**Theorem 2.** *The reverse diffusion process parameterized using the GPL distribution in Eq. (9) can model any distribution over $S_n$.*

## F  Results on TV Distances between Riffle Shuffles

**Proposition 3.** *Let $t \neq t'$ be positive integers. Then*

$$D_{\text{TV}}\left(q_{\text{RS}}^{(t)}, q_{\text{RS}}^{(t')}\right) = \frac{1}{2} \sum_{r=1}^{n} A_{n,r} \left| \frac{1}{2^{tn}} \binom{n + 2^t - r}{n} - \frac{1}{2^{t'n}} \binom{n + 2^{t'} - r}{n} \right|, \tag{13}$$

*and*

$$D_{\text{TV}}\left(q_{\text{RS}}^{(t)}, u\right) = \frac{1}{2} \sum_{r=1}^{n} A_{n,r} \left| \frac{1}{2^{tn}} \binom{n + 2^t - r}{n} - \frac{1}{n!} \right|. \tag{14}$$

*Proof.* Let $\sigma \in S_n$. It was shown in Bayer & Diaconis (1992) that

$$q_{\text{RS}}^{(t)}(\sigma) = \frac{1}{2^{tn}} \cdot \binom{n + 2^t - r}{n},$$

where $r$ is the number of rising sequences of $\sigma$. Note that if two permutations have the same number of rising sequences, then they have equal probability. Hence, we have

$$D_{\text{TV}}\left(q_{\text{RS}}^{(t)} - q_{\text{RS}}^{(t')}\right) = \frac{1}{2} \sum_{\sigma \in S_n} \left| q_{\text{RS}}^{(t)}(\sigma) - q_{\text{RS}}^{(t')}(\sigma) \right| = \frac{1}{2} \sum_{r=1}^{n} A_{n,r} \left| q_{\text{RS}}^{(t)}(\sigma) - q_{\text{RS}}^{(t')}(\sigma) \right|$$

$$= \frac{1}{2} \sum_{r=1}^{n} A_{n,r} \left| \frac{1}{2^{tn}} \binom{n + 2^t - r}{n} - \frac{1}{2^{t'n}} \binom{n + 2^{t'} - r}{n} \right|,$$

as claimed. For (14), replace $q_{\text{RS}}^{(t')}(\sigma)$ with $u(\sigma) = \frac{1}{n!}$ in the above derivations. $\qquad\square$

## G  Additional Details on Experiments

### G.1  Additional Synthetic Experiment: Learning a Single Permutation List

We present an additional synthetic experiment to illustrate the effectiveness of SymmetricDiffusers. A natural approach to modeling permutations is to represent them as sequences of numbers from $\{0, 1, \ldots, n-1\}$ and apply sequence generation models. However, existing sequence generation models relax the permutation constraint, allowing numbers to repeat and modeling transitions in a larger sequence space. To highlight the advantage, *i.e.*, restricting the diffusion trajectory to $S_n$ and modeling transitions in a smaller space ($O(n!)$ vs $O(n^n)$), of our approach, we conduct a synthetic experiment: learning a delta distribution over $S_n$, *i.e.*, a single permutation. In particular, we compare our method with SEDD (Lou et al., 2024), one of the strongest discrete diffusion models.

In the experiment, we train models to learn the identity permutation ($n = 100$) and a fixed arbitrary distribution ($n = 100$ and 200). For details on hyperparameters and training, please refer to Appendix G.5. From Table 5, we see that our method reaches 100% accuracy in all settings. While SEDD also performs well, the gap between its accuracy and ours grows with sequence length. These experiments verify the advantage and the effectiveness of our method.

We would also like to highlight another key limitation of other discrete diffusion models (including SEDD). In fact, it is nearly impossible for other discrete diffusion models to solve the tasks introduced in our paper, including the jigsaw puzzle, sorting multi-digit MNIST numbers, and the TSP. The

Table 5: Results on the toy experiment of learning a single permutation list.

| Method | Identity $n = 100$ | | Arbitrary Permutation $n = 100$ | | Arbitrary Permutation $n = 200$ | |
|---|---|---|---|---|---|---|
| | Accuracy (%) | Correct (%) | Accuracy (%) | Correct (%) | Accuracy (%) | Correct (%) |
| SEDD (Lou et al., 2024) | 95.47 | 99.95 | 93.24 | 99.93 | 88.75 | 99.94 |
| Ours | **100** | **100** | **100** | **100** | **100** | **100** |

reason is that all prior discrete diffusion models assume a fixed alphabet or vocabulary, and they model categorical distributions on the fixed alphabet. For example, in NLP tasks, the vocabulary is predefined and fixed. However, the alphabet is the set of all possible image patches for image tasks such as jigsaw puzzles and sorting MNIST numbers. It is impractical to gather the complete alphabet beforehand. We could potentially train VQVAEs to obtain quantized image embeddings. However, such an approach introduces the approximation error from the quantized alphabet. For the TSP, each node of the graph is a point in the continuous space $\mathbb{R}^2$, so it is also impossible to gather the complete alphabet. In contrast, our method can be successfully applied to these tasks because we model a distribution on the fixed alphabet $S_n$, and we treat each permutation as a function that can be applied to an ordered list of objects.

## G.2 DATASETS USED IN THE FULL PAPER

**Jigsaw Puzzle.** We created the Noisy MNIST dataset by adding *i.i.d.* Gaussian noise with a mean of 0 and a standard deviation of 0.01 to each pixel of the MNIST images. No noise was added to the CIFAR-10 images. The noisy images are then saved as the Noisy MNIST dataset. During training, each image is divided into $n \times n$ patches. A permutation is then sampled uniformly at random to shuffle these patches. The training set for Noisy MNIST comprises 60,000 images, while the CIFAR-10 training set contains 10,000 images. The Noisy MNIST test set, which is pre-shuffled, also includes 10,000 images. The CIFAR-10 test set, which shuffles images on the fly, contains 10,000 images as well.

**Sort 4-Digit MNIST Numbers.** For each training epoch, we generate 60,000 sequences of 4-digit MNIST images, each of length $n$, constructed dynamically on the fly. These 4-digit MNIST numbers are created by concatenating four MNIST images, each selected uniformly at random from the entire MNIST dataset, which consists of 60,000 images. For testing purposes, we similarly generate 10,000 sequences of $n$ 4-digit MNIST numbers on the fly.

**TSP.** We take the TSP-20 and TSP-50 dataset from Joshi et al. (2021) [1]. The train set consists of 1,512,000 graphs, where each node is an *i.i.d.* sample from the unit square $[0, 1]^2$. The labels are optimal TSP tours provided by the Concorde solver (Applegate et al., 2006). The test set consists of 1,280 graphs, with ground truth tour generated by the Concorde solver as well.

## G.3 ABLATION STUDIES

**Choices for Reverse Transition and Decoding Strategies.** As demonstrated in Table 6, we have explored various combinations of forward and inverse shuffling methods across tasks involving different sequence lengths. Both GPL and PL consistently excel in all experimental scenarios, highlighting their robustness and effectiveness. It is important to note that strategies such as random transposition and random insertion paired with their respective inverse operations, are less suitable for tasks with longer sequences. This limitation is attributed to the prolonged mixing times required by these two shuffling methods, a challenge that is thoroughly discussed in Section 3.1.2.

**Denoising Schedule.** We also conduct an ablation study on how we should merge reverse steps. As shown in Table 7, the choice of the denoising schedule can significantly affect the final performance. In particular, for $n = 100$ on the Sort 4-Digit MNIST Numbers task, the fact that $[0, 15]$ has 0 accuracy justifies our motivation to use diffusion to break down learning into smaller steps. The result we get also matches with our proposed heuristic in Section 3.4.

---

[1] `https://github.com/chaitjo/learning-tsp?tab=readme-ov-file`

Table 6: More results on sorting the 4-digit MNIST dataset using different combinations of forward process methods and reverse process methods. Results averaged over 3 runs with different seeds. RS: riffle shuffle; GPL: generalized Plackett-Luce; IRS: inverse riffle shuffle; RT: random transposition; IT: inverse transposition; RI: random insertion; II: inverse insertion.

| | | Sequence Length | | |
|---|---|---|---|---|
| | | 9 | 32 | 52 |
| RS (forward) + GPL (reverse) + greedy | Denoising Schedule | $[0, 3, 5, 9]$ | $[0, 5, 7, 12]$ | $[0, 5, 6, 7, 10, 13]$ |
| | Kendall-Tau $\uparrow$ | 0.948 | 0.857 | 0.779 |
| | Accuracy (%) | 89.4 | 54.8 | 24.4 |
| | Correct (%) | 95.9 | 88.1 | 81.6 |
| RS (forward) + PL (reverse) + greedy | Denoising Schedule | $[0, 3, 5, 9]$ | $[0, 5, 7, 12]$ | $[0, 5, 6, 7, 10, 13]$ |
| | Kendall-Tau | 0.953 | 0.867 | 0.799 |
| | Accuracy (%) | 90.9 | 56.4 | 26.4 |
| | Correct (%) | 96.4 | 89.0 | 83.3 |
| RS (forward) + PL (reverse) + beam search | Denoising Schedule | $[0, 3, 5, 9]$ | $[0, 5, 7, 12]$ | $[0, 5, 6, 7, 10, 13]$ |
| | Kendall-Tau $\uparrow$ | 0.955 | 0.869 | 0.797 |
| | Accuracy (%) | 91.1 | 57.2 | 26.4 |
| | Correct (%) | 96.5 | 89.2 | 83.1 |
| RS (forward) + IRS (reverse) + greedy | $T$ | 9 | 12 | 13 |
| | Kendall-Tau $\uparrow$ | 0.947 | 0.794 | 0.390 |
| | Accuracy (%) | 88.6 | 24.4 | 0.6 |
| | Correct (%) | 95.9 | 82.5 | 44.6 |
| RT (forward) + IT (reverse) + greedy | $T$ (using approx. $\frac{n}{2} \log n$) | 15 | 55 | 105 |
| | Kendall-Tau $\uparrow$ | 0.490 | | |
| | Accuracy (%) | 18.0 | Out of Memory | |
| | Correct (%) | 59.5 | | |
| RI (forward) + II (reverse) + greedy | $T$ (using approx. $n \log n$) | 25 | 110 | 205 |
| | Kendall-Tau $\uparrow$ | 0.954 | | |
| | Accuracy (%) | 91.1 | Out of Memory | |
| | Correct (%) | 96.4 | | |

Table 7: Results of sorting 100 4-digit MNIST images using various denoising schedules with the combination of RS, GPL and beam search consistently applied.

| Denoising Schedule | $[0, 15]$ | $[0, 8, 9, 15]$ | $[0, 7, 8, 9, 15]$ | $[0, 7, 8, 10, 15]$ | $[0, 8, 10, 15]$ |
|---|---|---|---|---|---|
| Kendall-Tau $\uparrow$ | 0.000 | 0.316 | 0.000 | 0.000 | **0.646** |
| Accuracy (%) | 0.0 | 0.0 | 0.0 | 0.0 | **4.5** |
| Correct (%) | 1.0 | 39.6 | 1.0 | 1.0 | **69.8** |

## G.4 LATENT LOSS IN JIGSAW PUZZLE

In the original setup of the Jigsaw Puzzle experiment using the Gumbel-Sinkhorn network (Mena et al., 2018), the permutations are latent. That is, the loss function in Gumbel-Sinkhorn is a pixel-level MSE loss and does not use the ground truth permutation label. However, our loss function (12) actually (implicitly) uses the ground truth permutation that maps the shuffled image patches to their original order. Therefore, for fair comparison with the Gumbel-Sinkhorn network in the Jigsaw Puzzle experiment, we modify our loss function so that it does not use the ground truth permutation. Recall from Section 3.2 that we defined

$$p_\theta(X_{t-1}|X_t) = \sum_{\sigma_t' \in \mathcal{T}} p(X_{t-1}|X_t, \sigma_t') p_\theta(\sigma_t'|X_t). \tag{22}$$

In our original setup, we defined $p(X_{t-1}|X_t, \sigma_t')$ as a delta distribution $\delta(X_{t-1} = Q_{\sigma_t'}X_t)$, but this would require that we know the permutation that turns $X_{t-1}$ to $X_t$, which is part of the ground truth. So instead, we parameterize $p(X_{t-1}|X_t, \sigma_t')$ as a Gaussian distribution $\mathcal{N}(X_{t-1}|Q_{\sigma_t}X_t, I)$. At the same time, we note that to find the gradient of (12), it suffices to find the gradient of the log of (22). We use the REINFORCE trick (Williams, 1992) to find the gradient of $\log p_\theta(X_{t-1}|X_t)$, which gives

us

$$\nabla_\theta \log p_\theta(X_{t-1}|X_t)$$

$$= \frac{1}{\sum\limits_{\sigma'_t \in \mathcal{T}} p(X_{t-1}|X_t, \sigma'_t)p_\theta(\sigma'_t|X_t)} \cdot \sum_{\sigma'_t \in \mathcal{T}} p(X_{t-1}|X_t, \sigma'_t)\nabla_\theta p_\theta(\sigma'_t|X_t)$$

$$= \frac{1}{\sum\limits_{\sigma'_t \in \mathcal{T}} p(X_{t-1}|X_t, \sigma'_t)p_\theta(\sigma'_t|X_t)} \cdot \sum_{\sigma'_t \in \mathcal{T}} p(X_{t-1}|X_t, \sigma'_t)p_\theta(\sigma'_t|X_t)\big(\nabla_\theta \log p_\theta(\sigma_t|X_t)\big)$$

$$= \frac{\mathbb{E}_{p_\theta(\sigma_t|X_t)}\Big[p(X_{t-1}|X_t, \sigma'_t)\nabla_\theta \log p_\theta(\sigma_t|X_t)\Big]}{\mathbb{E}_{p_\theta(\sigma_t|X_t)}\Big[p(X_{t-1}|X_t, \sigma'_t)\Big]}$$

$$\approx \sum_{n=1}^{N} \frac{p\left(X_{t-1}|X_t, \sigma_t^{(n)}\right)}{\sum_{m=1}^{N} p\left(X_{t-1}|X_t, \sigma_t^{(m)}\right)} \cdot \nabla_\theta \log p_\theta\left(\sigma_t^{(n)}|X_t\right),$$

where we have used Monte-Carlo estimation in the last step, and $\sigma_t^{(1)}, \ldots, \sigma_t^{(N)} \sim p_\theta(\sigma_t|X_t)$. We further add an entropy regularization term $-\lambda \cdot \mathbb{E}_{p_\theta(\sigma_t|X_t)}[\log p_\theta(\sigma_t|X_t)]$ to each of $\log p_\theta(X_{t-1}|X_t)$. Using the same REINFORCE and Monte-Carlo trick, we obtain

$$\nabla_\theta\left(-\lambda \cdot \mathbb{E}_{p_\theta(\sigma_t|X_t)}\Big[\log p_\theta(\sigma_t|X_t)\Big]\right) \approx \sum_{n=1}^{N} -\lambda \log p_\theta\left(\sigma_t^{(n)}|X_t\right)\nabla_\theta \log p_\theta\left(\sigma_t^{(n)}|X_t\right),$$

where $\sigma_t^{(1)}, \ldots, \sigma_t^{(N)} \sim p_\theta(\sigma_t|X_t)$. Therefore, we have

$$\nabla_\theta\left(\log p_\theta(X_{t-1}|X_t) - \lambda \cdot \mathbb{E}_{p_\theta(\sigma_t|X_t)}\Big[\log p_\theta(\sigma_t|X_t)\Big]\right)$$

$$\approx \sum_{n=1}^{N}\left(\underbrace{\frac{p\left(X_{t-1}|X_t, \sigma_t^{(n)}\right)}{\sum_{m=1}^{N} p\left(X_{t-1}|X_t, \sigma_t^{(m)}\right)} - \lambda \log p_\theta\left(\sigma_t^{(n)}|X_t\right)}_{\texttt{weight}}\right) \cdot \nabla_\theta \log p_\theta\left(\sigma_t^{(n)}|X_t\right), \quad (23)$$

where $\sigma_t^{(1)}, \ldots, \sigma_t^{(N)} \sim p_\theta(\sigma_t|X_t)$. We then substitute in

$$p\left(X_{t-1}|X_t, \sigma_t^{(n)}\right) = \mathcal{N}\left(X_{t-1}|Q_{\sigma_t^{(n)}}X_t, I\right)$$

for all $n \in [N]$. Finally, we also subtract the exponential moving average `weight` as a control variate for variance reduction, where the exponential moving average is given by `ema` $\leftarrow$ `ema_rate` $\cdot$ `ema` $+ (1 - $ `ema_rate`$) \cdot$ `weight` for each gradient descent step.

## G.5 TRAINING DETAILS AND ARCHITECTURE HYPERPARAMETERS

**Hardware.** The Jigsaw Puzzle and Sort 4-Digit MNIST Numbers experiments are trained and evaluated on the NVIDIA A40 GPU. The TSP experiments are trained and evaluated on the NVIDIA A40 and A100 GPU.

**Learning a Single Permutation List.** The SEDD model we use in our experiments has about 25M parameters. We use 7 encoder layers, 8 heads, model dimension 512, feed-forward dimension 2048, and dropout 0.1. We use the uniform transition for the forward process following the original work.

Our model only has about 2M parameters. We use 7 encoder layers, 8 heads, model dimension 128, feed-forward dimension 512, and dropout 0.1. We also use a learned embedding for the numbers. For the diffusion process, we use a denoising schedule of $[0, 8, 10, 15]$ for $n = 100$ and $[0, 9, 10, 12]$ for $n = 200$.

For $n = 100$, we use a batch size of 512 and 30K training steps on both methods. For $n = 200$, we use a batch size of 128 and 30K training steps on both methods. For performance evaluation, we randomly sample 2560 sequences for SEDD and 2560 permutations for our method, and we perform their respective decoding processes.

**Jigsaw Puzzle.**    For the Jigsaw Puzzle experiments, we use the AdamW optimizer (Loshchilov & Hutter, 2019) with weight decay 1e-2, $\varepsilon = $ 1e-9, and $\beta = (0.9, 0.98)$. We use the Noam learning rate scheduler given in (Vaswani et al., 2023) with 51,600 warmup steps for Noisy MNIST and 46,000 steps for CIFAR-10. We train for 120 epochs with a batch size of 64. When computing the loss (12), we use Monte-Carlo estimation for the expectation and sample 3 trajectories. For REINFORCE, we sampled 10 times for the Monte-Carlo estimation in (23), and we used an entropy regularization rate $\lambda = 0.05$ and an `ema_rate` of 0.995. The neural network architecture and related hyperparameters are given in Table 8. The denoising schedules, with riffle shuffles as the forward process and GPL as the reverse process, are give in Table 9. For beam search, we use a beam size of 200 when decoding from GPL, and we use a beam size of 20 when decoding along the diffusion denoising schedule.

Table 8: Jigsaw puzzle neural network architecture and hyperparameters.

| Layer | Details |
|---|---|
| Convolution | Output channels 32, kernel size 3, padding 1, stride 1 |
| Batch Normalization | − |
| ReLU | − |
| Max-pooling | Pooling 2 |
| Fully-connected | Output dimension (`dim_after_conv` $+ 128)/2$ |
| ReLU | − |
| Fully-connected | Output dimension 128 |
| Transformer encoder | 7 layers, 8 heads, model dimension ($d_{\text{model}}$) 128, feed-forward dimension 512, dropout 0.1 |

Table 9: Denoising schedules for the Jigsaw Puzzle task, where we use riffle shuffle in the forward process and GPL in the revserse process.

| Number of patches per side | Denoising schedule |
|---|---|
| $2 \times 2$ | $[0, 2, 7]$ |
| $3 \times 3$ | $[0, 3, 5, 9]$ |
| $4 \times 4$ | $[0, 4, 6, 10]$ |
| $5 \times 5$ | $[0, 5, 7, 11]$ |
| $6 \times 6$ | $[0, 6, 8, 12]$ |

**Sort 4-Digit MNIST Numbers.**    For the task of sorting 4-digit MNIST numbers with $n \leq 100$, we use the exact training and beam search setup as the Jigsaw Puzzle, except that we do not need to use REINFORCE. The neural network architecture is given in Table 10, The denoising schedules, with riffle shuffles as the forward process and GPL as the reverse process, are give in Table 11.

For $n = 200$, we use the cosine decay learning rate schedule with 2350 steps of linear warmup and maximum learning rate 5e-5. The neural network architecture is the same as that of $n \leq 100$, with the exception that we use $d_{\text{model}} = d_{\text{feed-forward}} = 768$, 12 layers, and 12 heads for the transformer encoder layer. We use the PL distribution for the reverse process. When computing the loss, we randomly sample a timestep from the denoising schedule as in Eq.(20) in Appendix C.2 due to efficiency reasons. All other setups are the same as that of $n \leq 100$.

**TSP.**    For solving the TSP, we perform supervised learning to train our SymmetricDiffusers to solve the TSP. Let $\sigma^*$ be an optimal permutation, and let $X_0$ be the list of nodes ordered by $\sigma^*$. We note that any cyclic shift of $X_0$ is also optimal. Thus, for simplicity and without loss of generality, we always assume $\sigma^*(1) = 1$. In the forward process of SymmetricDiffusers, we only shuffle the second to the $n^{\text{th}}$ node (or component). In the reverse process, we mask certain parameters of the reverse distribution so that we will always sample a permutation with $\sigma_t(1) = 1$.

The architecture details are slightly different for TSP-20, where we input both node and edge features into our network. Denote by $X_t$ the ordered list of nodes at time $t$. We obtain $Y_t \in \mathbb{R}^{n \times d_{\text{model}}}$ as in Eq. (15), where `encoder`$_\theta$ is now a sinusoidal embedding of the 2D coordinates. Let $D_t \in \mathbb{R}^{n \times n}$ be the

Table 10: Sort 4-digit MNIST numbers neural network architecture and hyperparameters.

| Layer | Details |
|---|---|
| Convolution | Output channels 32, kernel size 5, padding 2, stride 1 |
| Batch Normalization | – |
| ReLU | – |
| Max-pooling | Pooling 2 |
| Convolution | Output channels 64, kernel size 5, padding 2, stride 1 |
| Batch Normalization | – |
| ReLU | – |
| Max-pooling | Pooling 2 |
| Fully-connected | Output dimension $(\texttt{dim\_after\_conv} + 128)/2$ |
| ReLU | – |
| Fully-connected | Output dimension 128 |
| Transformer encoder | 7 layers, 8 heads, model dimension ($d_{\mathrm{model}}$) 128, feed-forward dimension 512, dropout 0.1 |

Table 11: Denoising schedules for the Sort 4-Digit MNIST Numbers task, where we use riffle shuffle in the forward process and GPL in the revserse process.

| Sequence Length $n$ | Denoising schedule |
|---|---|
| 3 | $[0, 2, 7]$ |
| 5 | $[0, 2, 8]$ |
| 7 | $[0, 3, 8]$ |
| 9 | $[0, 3, 5, 9]$ |
| 15 | $[0, 4, 7, 10]$ |
| 32 | $[0, 5, 7, 12]$ |
| 52 | $[0, 5, 6, 7, 10, 13]$ |
| 100 | $[0, 8, 10, 15]$ |
| 200 | $[0, 9, 10, 12]$ |

matrix representing the pairwise distances of points in $X_t$, respecting the order in $X_t$. Let $E_t \in \mathbb{R}^{\binom{n}{2}}$ be the flattened vector of the upper triangular part of $D_t$. We also apply sinusoidal embedding to $E_t$ and add $\texttt{time\_embd}(t)$ to it. We call the result $F_t \in \mathbb{R}^{\binom{n}{2} \times d_{\mathrm{model}}}$.

Now, instead of applying the usual transformer encoder with self-attentions, we alternate between cross-attentions and self-attentions. For cross-attention layers, we use the node representations from the previous layer as the query, and we always use $K = V = F_t$. We also apply an attention mask to the cross-attention, so that each node will only attend to edges that it is incident with. For self-attention layers, we always use the node representations from the previous layer as input. We always use an even number of layers, with the first layer being a cross-attention layer, and the last layer being a self-attention layer structured to produce the required parameters for the reverse distribution as illustrated in Appendix B. For hyperparameters, we use 16 alternating layers, 8 attention heads, $d_{\mathrm{model}} = 256$, feed-forward hidden dimension 1024, and dropout rate 0.1.

For training details on the TSP-20 task, we use the AdamW optimizer (Loshchilov & Hutter, 2019) with weight decay 1e-4, $\varepsilon = $ 1e-8, and $\beta = (0.9, 0.999)$. We use the cosine annealing learning rate scheduler starting from 2e-4 and ending at 0. We train for 50 epochs with a batch size of 512. When computing the loss (12), we use Monte-Carlo estimation for the expectation and sample 1 trajectory. We use a denoising schedule of $[0, 4, 5, 7]$, with riffle shuffles as the forward process and GPL as the reverse process. Finally, we use beam search for decoding, and we use a beam size of 256 both when decoding from GPL and decoding along the denoising schedule.

For the TSP-50 task, we use the original architecture stated in Appendix B without the edge information. We train for 250 epochs with a batch size of 256. We use a denoising schedule of $[0, 5, 6, 7]$. We use a beam size of 768 when decoding from GPL and decoding along the denoising schedule.

When decoding, we pick the permutation that gives the least TSP tour length in the final step of the beam search. All other setups are identical with that of TSP-20.

## G.6 BASELINES IMPLEMENTATION DETAILS

**Gumbel-Sinkhorn Network.** We have re-implemented the Gumbel-Sinkhorn Network (Mena et al., 2018) for application on jigsaw puzzles, following the implementations provided in the official repository[2]. To ensure a fair comparison, we conducted a thorough grid search of the model's hyper-parameters. The parameters included in our search space are as follows,

Table 12: Hyperparameter Search Space for the Gumbel-Sinkhorn Network

| Hyperparameter | Values |
|---|---|
| Learning Rate (lr) | $\{10^{-3}, 10^{-4}, 10^{-5}\}$ |
| Batch Size | $\{50\}$ |
| Hidden Channels | $\{64, 128\}$ |
| Kernel Size | $\{3, 5\}$ |
| $\tau$ | $\{0.2, 0.5, 1, 2, 5\}$ |
| Number of Sinkhorn Iterations (n_sink_iter) | $\{20\}$ |
| Number of Samples | $\{10\}$ |

**Diffsort & Error-free Diffsort** We have implemented two differentiable sorting networks from the official repository[3] specific to error-free diffsort. For sorting 4-digit MNIST images, error-free diffsort employs TransformerL as its backbone, with detailed hyperparameters listed in Table 13. Conversely, Diffsort uses a CNN as its backbone, with a learning rate set to $10^{-3.5}$; the relevant hyperparameters are outlined in Table 14.

For jigsaw puzzle tasks, error-free diffsort continues to utilize a transformer, whereas Diffsort employs a CNN. For other configurations, we align the settings with those of tasks having similar sequence lengths in the 4-digit MNIST sorting task. For instance, for $3 \times 3$ puzzles, we apply the same configuration as used for sorting tasks with a sequence length of 9.

Table 13: Hyperparameters for Error-Free Diffsort on Sorting 4-Digit MNIST Numbers

| Sequence Length | Steepness | Sorting Network | Loss Weight | Learning Rate |
|---|---|---|---|---|
| 3 | 10 | odd even | 1.00 | $10^{-4}$ |
| 5 | 26 | odd even | 1.00 | $10^{-4}$ |
| 7 | 31 | odd even | 1.00 | $10^{-4}$ |
| 9 | 34 | odd even | 1.00 | $10^{-4}$ |
| 15 | 25 | odd even | 0.10 | $10^{-4}$ |
| 32 | 124 | odd even | 0.10 | $10^{-4}$ |
| 52 | 130 | bitonic | 0.10 | $10^{-3.5}$ |
| 100 | 140 | bitonic | 0.10 | $10^{-3.5}$ |
| 200 | 200 | bitonic | 0.10 | $10^{-4}$ |

**TSP.** For the baselines for TSP, we first have 4 traditional operations research solvers. Gurobi (Gurobi Optimization, LLC, 2023) and Concorde (Applegate et al., 2006) are known as exact solvers, while LKH-3 (Helsgaun, 2017) is a strong heuristic and 2-Opt (Lin & Kernighan, 1973) is a weak heuristic. For LKH-3, we used 500 trials, and for 2-Opt, we used 5 random initial guesses with seed 42.

For the GCN model (Joshi et al., 2019), we utilized the official repository[4] and adhered closely to its default configuration for the TSP-20 dataset. For DIFUSCO (Sun & Yang, 2023), we sourced it from

---

[2]https://github.com/google/gumbel_sinkhorn
[3]https://github.com/jungtaekkim/error-free-differentiable-swap-functions
[4]https://github.com/chaitjo/graph-convnet-tsp

Table 14: Hyperparameters for Diffsort on Sorting 4-Digit MNIST Numbers

| Sequence Length | Steepness | Sorting Network |
|:---:|:---:|:---:|
| 3 | 6 | odd even |
| 5 | 20 | odd even |
| 7 | 29 | odd even |
| 9 | 32 | odd even |
| 15 | 25 | odd even |
| 32 | 25 | bitonic |
| 52 | 25 | bitonic |
| 100 | 25 | bitonic |
| 200 | 200 | bitonic |

its official repository[5] and followed the recommended configuration of TSP-50 dataset, with a minor adjustment in the batch size. We increased the batch size to 512 to accelerate the training process. For fair comparison, we also remove the post-processing heuristics in both models during the evaluation.

## H LIMITATIONS

Despite the success of this method on various tasks, the model presented in this paper still requires a time-space complexity of $O(n^2)$ due to its reliance on the parametric representation of GPL and the backbone of transformer attention layers. This complexity poses a significant challenge in scaling up to applications involving larger symmetric groups or Lie groups.

---

[5]https://github.com/Edward-Sun/DIFUSCO

