# OpenReview forum: "SymmetricDiffusers: Learning Discrete Diffusion on Finite Symmetric Groups"
_ICLR.cc/2025/Conference — ICLR 2025 Oral_

### Official Review · Reviewer_hmq7 · 2024-10-25

**Soundness:** 3
**Presentation:** 3
**Contribution:** 3
**Rating:** 8
**Confidence:** 3

**Summary:**

This authors proposes a diffusion framework that learns a distribution over permutations.
They propose a set of different reverse and forward diffusion processes to realize this goal.
They validate their approach on various different experiments such as TSP, four-digit MNIST, CIFAR and noisy MNIST.

**Strengths:**

1. The paper presents an interesting approach with mostly clear writing
2. Several innovative reverse and forward processes are proposed

**Weaknesses:**

1. The claim of state-of-the-art performance on TSP-20 appears too bold given the limited scope
2. Several experimental details lack sufficient clarity (see Questions)

**Questions:**

1. How do the proposed reverse and forward processes compare to naive discrete denoising diffusion models?
What is the difference between your method and other differentiable sorting baselines in Tab. 1?

2. What is the sequence length $n$ in the four-digit MNIST dataset?

3. Why are experiments limited to TSP-20 instances rather than including larger TSP instances? TSP-20 is considered a very small problem size. This experiment is the most interesting to me and it would be interesting to see how this method performs on larger TSP instances. Can you provide an additional comparison on TSP-100?

4. Can your method be combined with the framework proposed in [1] to solve TSP using diffusion models without requiring data from classical solvers?



## References
[1] Sanokowski, Sebastian, Sepp Hochreiter, and Sebastian Lehner. "A Diffusion Model Framework for Unsupervised Neural Combinatorial Optimization." Forty-First International Conference on Machine Learning.

---

> ### Author Response · Authors · 2024-11-20
> **Response to Reviewer hmq7 (Part 1/2)**
>
> Thank you for the insightful and constructive comments. We appreciate your positive feedback and address the questions below.
>
> > **Q1:** How do the proposed reverse and forward processes compare to naive discrete denoising diffusion models? What is the difference between your method and other differentiable sorting baselines in Tab. 1?
> >
>
> **A1:** Existing discrete diffusion models assume token-wise conditional independence when modeling the reverse transition distributions. This assumption does not hold in learning permutations since different components of $X_{t-1}$ are **not** **independent** conditioned on $X_t$ in the reverse process, and they have to satisfy the constraint of permutations (i.e., being one of the vertices of Birkhoff Polytope). Therefore, if we have a distribution over $[n]^n$, the denoising step in standard diffusion models would lead to noisy data $X_{t-1}$ that is not an exact permutation. Furthermore, it is also computationally expensive to project it to a distribution over $S_n$.
>
> Discrete diffusion methods like D3PM [2], which model categorical distributions, are also unsuitable for $S_n$. These methods require explicit matrix multiplications involving $n!\times n!$ transition matrices. While D3PM uses dense transition matrices such as uniform or discretized Gaussian distributions, performing dense matrix multiplications at this scale is impractical.
>
> Our proposed method addresses these challenges by defining efficient, customized transition distributions through card-shuffling methods. This approach avoids explicit matrix multiplications by directly simulating the forward process using the efficient operations of shuffling methods. Essentially, the shuffling methods induce “sparse” transitions on $S_n$, resolving the efficiency issues inherent in existing discrete diffusion models. As our framework is fundamentally different and existing frameworks are infeasible for $S_n$, our baselines focus on comparing different shuffling methods within our framework.
>
> The differentiable sorting baselines in Table 1 represent the predominant approach in the literature for learning permutations. They define differentiable approximations to sorting operations, enabling optimization over permutations. In contrast, our method uses discrete diffusion models to learn a distribution over $S_n$ via forward noising and reverse denoising processes—a fundamentally different paradigm from differentiable sorting. Additionally, our method significantly outperforms differentiable sorting methods on longer sequence lengths, marking a substantial improvement in permutation learning.
>
> > **Q2:** What is the sequence length $n$ in the four-digit MNIST dataset?
> >
>
> **A2:** The sequence length $n$ is the number of four-digit MNIST numbers that we are sorting, which is also the $n$ in $S_n$. We have clarified this on line 423 **(colored blue)** of the newly uploaded paper.
>
> ## References
>
> [1] Sanokowski, Sebastian, Sepp Hochreiter, and Sebastian Lehner. "A Diffusion Model Framework for Unsupervised Neural Combinatorial Optimization." Forty-First International Conference on Machine Learning.
>
> [2] Austin et al. "Structured denoising diffusion models in discrete state-spaces." Advances in Neural Information Processing Systems 34 (2021): 17981-17993.

---

> ### Author Response · Authors · 2024-11-20
> **Response to Reviewer hmq7 (Part 2/2)**
>
> > **Q3:** Why are experiments limited to TSP-20 instances rather than including larger TSP instances? TSP-20 is considered a very small problem size. This experiment is the most interesting to me and it would be interesting to see how this method performs on larger TSP instances. Can you provide an additional comparison on TSP-100?
> >
>
> **A3:** We are running larger-scale TSP experiments like TSP-50 and TSP-100 right now, but it takes much longer to train the models compared to other experiments. Since there is only limited time for the rebuttal period, we will post a follow-up response if our experiments finish before the discussion deadline.
>
> We would like to point out that the current literature on permutation learning has never done experiments on TSPs. As a general permutation learning model, our model is certainly different from models specifically designed for the TSP. The TSP is just one application, and our model has been verified to be much more effective than previous methods on other tasks like the jigsaw puzzle and sorting 4-digit MNIST numbers. Previous work, such as the baseline methods in our experiments, can only learn permutations up to a small sequence length. For example, for the sort 4-digit MNIST numbers experiment, previous methods are only effective for sequences of length up to $32$, while our method has promising results for sequence lengths up to $200$ and outperforms the baseline methods significantly under longer sequence lengths. Our method already provide a substantial improvement over previous methods in the field of permutation learning.
>
> > **Q4:** Can your method be combined with the framework proposed in [1] to solve TSP using diffusion models without requiring data from classical solvers?
> >
>
> **A4:** Thanks for pointing out the reference! The framework proposed in [1] offers a compelling approach to unsupervised neural combinatorial optimization. In particular, [1] uses an energy $H$ as signals to the quality of a solution, and use a tractable Joint Variational Upper Bound of the commonly used reverse KL divergence to bypass the need for exact likelihood evaluations. Diffusion models can be naturally incorporated into the Joint Variational Upper Bound. Specifically, our method can certainly be combined with the framework in [1] as long as we pick a reverse transition such that the Shannon entropy $S(q_{\theta}(X_{t-1}|X_t))$ in Eq.(6) of [1] is tractable. Solving the TSP without requiring data from classical solvers is definitely an exiciting research direction, and combining our method and [1] could be an interesting future work. **We have discussed [1] in the newly uploaded version of our paper (line 517 on page 10, colored blue).**
>
> ## References
>
> [1] Sanokowski, Sebastian, Sepp Hochreiter, and Sebastian Lehner. "A Diffusion Model Framework for Unsupervised Neural Combinatorial Optimization." Forty-First International Conference on Machine Learning.
>
> [2] Austin et al. "Structured denoising diffusion models in discrete state-spaces." Advances in Neural Information Processing Systems 34 (2021): 17981-17993.

---

> ### Comment · Reviewer_hmq7 · 2024-11-22
> **Raised Score**
>
> I thank the authors for their detailed answers and I have raised my score to 8.
> I am glad to hear that the authors are conducting further experiments on TSP and I am curious to see what comes out in the large-scale TSP experiments!
>
> I have a follow-up question:
> You are writing that the framework from Sanokowski et al. 2024 can be used "as long as we pick a reverse transition such that the Shannon entropy $S(q_\theta(X_{t-1}|X_t))$ is tractable".
> What do you exactly mean by that? I see that in Sanokowski et al. 2024  $q_\theta(X_{t-1}|X_t)$ is chosen so that $S(q_\theta(X_{t-1}|X_t))$ can be calculated exactly. But it is not a strict requirement as you can alternatively Monte Carlo estimate the entropy with $S(q_\theta(X_{t-1}|X_t)) = -  E_{X_{t-1} \sim q_\theta(X_{t-1}|X_t)}  [ \log q_\theta(X_{t-1}|X_t)  ]$, which is possible as long as $q_\theta(X_{t-1}|X_t)$ can be evaluated.
> So my question reduces to whether your proposed reverse diffusion transitions $q_\theta(X_{t-1}|X_t)$ can be evaluated?
>
> For me what you write down in L. 350 ff. "Note that although we may not have the analytical form of $q(X_t|X_{t-1})$, we can draw samples from it." would rather be a show stopper of the framework from Sanokowski et al. 2024, because $q(X_t|X_{t-1})$ must be analytically available.

---

> > ### Author Response · Authors · 2024-11-25
> > **Thank you and follow-up**
> >
> > Thank you for your reply and positive feedback! We appreciate your time in reviewing our paper and responses!
> >
> > You are correct that we can use a Monte Carlo estimation of the entropy $S(q\_{\theta}(X\_{t-1}|X\_t))=-\mathbb{E}\_{X\_{t-1}\sim q\_{\theta}(X\_{t-1}|X\_t)}\big[\log q\_{\theta}(X\_{t-1}|X\_t)\big]$, and the log likelihood $\log q_{\theta}(X_{t-1}|X_t)$ for PL or GPL is analytically available. Our original concern was that in Eq.(6) of [1], the term involving the entropy is $\mathbb{E}\_{X_{T:t}\sim q\_{\theta}(X\_{T:t})}\big[S(q\_{\theta}(X\_{t-1}|X\_t))\big]$, which already requires a Monte Carlo estimation when using REINFORCE to estimate the gradient. This means we have to use MC estimation twice, so there may be a very high variance.
> >
> > You bring up a good point that [1] also requires the forward transition $q(X_t|X_{t-1})$ to be analytically available, and it is true that most shuffling methods don’t admit analytical forms of $q(X_t|X_{t-1})$. However, for riffle shuffles, $q(X_t|X_{t-1})$ is actually available, which is briefly mentioned in Appendix C.1 line 774-775 of our paper. So despite the concern with high variance, it would be an interesting experiment to try the method in [1] using riffle shuffles.
> >
> > We now have some preliminary results on further experiments including the ImageNet jigsaw puzzle and TSP-50 experiments. For the ImageNet jigsaw puzzle experiments, due to time constraints, we ran on only the first 20 classes of the ImageNet dataset, and we currently only have results for $2\times 2$ and $3\times 3$. In the table below, the first two rows are the results of our method, and the last two rows are results from the Gumbel-Sinkhorn network. It is clear that our model outperforms the Gumbel-Sinkhorn network.
> >
> > |  |  | **Kendall-Tau $\uparrow$** | **Accuracy (%)** | **Correct (%)** | **RMSE** $\downarrow$ | **MAE** $\downarrow$ |
> > | --- | --- | --- | --- | --- | --- | --- |
> > | **SymmetricDiffusers (Ours)** | **$\mathbf{2\times 2}$** | **0.8627** | **83.50** | **89.88** | 0.1806 | **0.0413** |
> > |  | **$\mathbf{3\times 3}$** | **0.7451** | **57.50** | **79.31** | **0.2245** | **0.0687** |
> > |  |  |  |  |  |  |  |
> > | **Gumbel-Sinkhorn Network** | **$\mathbf{2\times 2}$** | 0.8212 | 78.36 | 86.26 | **0.1583** | 0.0687 |
> > |  | **$\mathbf{3\times 3}$** | 0.5667 | 19.10 | 55.83 | 0.3388 | 0.1636 |
> >
> > For the TSP-50 experiment, in the table below, Concorde and 2-Opt are OR solvers, and GCN, DIFUSCO, and Ours are learning-based models. Due to time constraints, we trained on only 1/3 of the training set (500K graphs), and the numbers for 2-Opt, GCN, and DIFUSCO are directly copied from the DIFUSCO paper [2]. The decoding heuristics used for GCN and DIFUSCO are greedy. The decoding heuristics used for our method are beam search and picking the tour with the shortest length in the final beam. Although our current results do not surpass the state-of-the-art, they are still comparable and demonstrate significant promise. As our method is not specifically tailored for TSPs, further hyperparameter tuning and architectural tweaking are required. Importantly, such modifications would not impact our core contribution, i.e., the discrete diffusion framework over finite symmetric groups. We plan to continue exploring this direction and will provide updated performance results on large-scale TSPs in a future version of our paper.
> >
> > | **Method** | **Concorde** | **2-Opt** | **GCN** | **DIFUSCO** | **Ours** |
> > | --- | --- | --- | --- | --- | --- |
> > | **Tour Length $\downarrow$** | **5.69** | 5.86 | 5.87 | **5.70** | 5.86 |
> > | **Optimality Gap (%) $\downarrow$** | **0.00** | 2.95 | 3.10 | **0.10** | 2.94 |
> >
> > ### References
> >
> > [1] Sanokowski, Sebastian, Sepp Hochreiter, and Sebastian Lehner. "A Diffusion Model Framework for Unsupervised Neural Combinatorial Optimization." Forty-First International Conference on Machine Learning.
> >
> > [2] Zhiqing Sun and Yiming Yang. Difusco: Graph-based diffusion solvers for combinatorial optimization, 2023.

---

### Official Review · Reviewer_iCzn · 2024-11-01

**Soundness:** 3
**Presentation:** 3
**Contribution:** 3
**Rating:** 8
**Confidence:** 4

**Summary:**

The paper studies the important question of sampling from the finite symmetric group $S_n$ via diffusion models. This is structurally different from prior works on diffusion models which concentrate on sampling from product spaces (i.e, sampling without replacement). The authors clearly introduce various noising Markov chains for sampling a uniform distribution over $S_n$ and show how to parametrize the forward and reverse processes. This is the main technical contribution of the paper. The training loss is then based the variational lower bound of D3PM (Austin et al). The paper achieves strong performance in solving jigsaw puzzles of MNIST and CIFAR 10, sorting numbers from MNIST and solving TSP problems.

**Strengths:**

The paper is clearly written and the Markov chains used to obtain uniformly random permutations are clearly surveyed. The construction of the parameterization of the reverse process and the loss function is nice and the algorithm outperforms existing methods in the empirical tasks.

**Weaknesses:**

1. Experiments are very small scale, comprising of sorting 4 digit MNIST, solving 20 node TSPs and solving jigsaw puzzles of CIFAR-10 data.

2. There is no substantive theoretical contribution other than introducing the parametrization for the reverse processes.

3. The reverse process for random transposition is not very expressive. Suppose the reverse transposition is $(1,2)$ with probability $0.5$ and $(2,3)$ with probability $0.5$. This simple distribution cannot be expressed using the model.

4. Note that $S_n \subseteq [n]^n$. If the input data comprises of only permutations, then the network should learn to sample from a distribution whose samples are permutations and the standard framework of diffusion models applies. This simple baseline has not been considered.

[Q related to 4] The authors also mention that representing a transition matrix over $S_n$ requires $n!\times n!$ sized matrix. However, the authors themselves give a succinct description/ representation of the forward transition matrix in the paper. The authors should elaborate why it is not possible to use this representation algorithmically.

**Minor:**
In proposition 1, should it be changed to "the GPL distribution can represent a delta distributions in the limit" instead of "exactly"?

*additional references:*
[1] Generating a random permutation with random transpositions by Diaconis and Shahshahani
[2] Simplified and Generalized Masked Diffusion for Discrete Data by Shi et al.
[3] Glauber Generative Model: Discrete Diffusion Models via Binary Classification by Varma et al.

**Questions:**

Address the points raised in the weaknesses section.

---

> ### Author Response · Authors · 2024-11-20
> **Response to Reviewer iCzn (Part 1/2)**
>
> Thank you for the insightful and constructive comments. We appreciate your positive feedback and address the questions below.
>
> > **Q1:** Experiments are very small scale, comprising of sorting 4 digit MNIST, solving 20 node TSPs and solving jigsaw puzzles of CIFAR-10 data.
> >
>
> **A1:**  We acknowledge the concern about the experiment scale. However, scalability remains a highly challenging aspect in permutation learning. Previous methods in the literature are generally limited to smaller permutation lengths. For example, for the sort 4-digit MNIST numbers experiment, previous methods are only effective for sequences of length up to $32$, while our method has promising results for sequence lengths up to $200$ and beats the baseline methods significantly in these longer sequence lengths. Furthermore, none of the previous permutation learning methods have tackled TSP experiments. While our model is not specifically designed for TSP tasks, it represents a substantial improvement over existing general-purpose permutation learning methods.
>
> To test our framework on more complicated and larger tasks, we are conducting additional experiments, including solving jigsaw puzzles on ImageNet and larger-scale TSP tasks (e.g., TSP-50). However, these experiments require extensive hyperparameter tuning and significantly longer training times, particularly for larger TSP instances. Due to the limited time during the rebuttal period, we will provide a follow-up response if these experiments complete before the discussion deadline.
>
> > **Q2:** The reverse process for random transposition is not very expressive. Suppose the reverse transposition is $(1,2)$ with probability $0.5$ and $(2,3)$ with probability $0.5$. This simple distribution cannot be expressed using the model.
> >
>
> **A2:** It is true that the inverse transposition model cannot represent the specific distribution you described. However, inverse transposition is only one of the many reverse process methods we proposed. Importantly, we demonstrate that the GPL distribution, which is the best-performer in our experiments, can approximate your distribution to any precision. The distribution you mentioned on $S_3$ is
>
> $$
> p\\left(\\begin{pmatrix}1&2&3\\\\2&1&3\\end{pmatrix}\\right)=p\\left(\\begin{pmatrix}1&2&3\\\\1&3&2\\end{pmatrix}\\right)=\\frac12.
> $$
>
> Consider the following score parameters for GPL
>
> $$
> S=\\begin{bmatrix}0&0&-\\infty\\\\0&-\\infty&-C\\\\-\\infty&0&0\\end{bmatrix},
> $$
>
> where $C$ is some arbitrary large positive number. Then we have
>
> $$
> \\begin{align*}
> \\text{GPL}\_S \\left( \\begin{pmatrix} 1&2&3 \\\\ 2&1&3 \\end{pmatrix} \\right) &= \\frac{\\exp(s_{12})}{\\exp(s_{11})+\\exp(s_{12})+\\exp(s_{13})}\\cdot\\frac{\\exp(s_{21})}{\\exp(s_{21})+\\exp(s_{23})}\\cdot\\frac{\\exp(s_{33})}{\\exp(s_{33})} \\\\
> &=\\frac{1}{1+1}\\cdot\\frac{1}{1+\\exp(-C)}\\to\\frac12
> \\end{align*}
> $$
>
> as $C\\to\\infty$. We also have
>
> $$
> \\begin{align*}
> \\text{GPL}\_S\\left(\\begin{pmatrix}1&2&3\\\\1&3&2\\end{pmatrix}\\right)&=\\frac{\\exp(s_{11})}{\\exp(s_{11})+\\exp(s_{12})+\\exp(s_{13})}\\cdot\\frac{\\exp(s_{23})}{\\exp(s_{22})+\\exp(s_{23})}\\cdot\\frac{\\exp(s_{32})}{\\exp(s_{32})} \\\\
> &=\\frac{1}{1+1}\\cdot\\frac{\\exp(-C)}{\\exp(-C)}\\cdot 1=\\frac12.
> \\end{align*}
> $$
>
> In fact, during the rebuttal period, we proved that the reverse process using GPL is expressive enough to represent **any** distribution over $S_n$, which is a significant result regarding the expressiveness of the reverse process. Please refer to Theorem 2 **(colored blue)** and Appendix E of the newly uploaded paper. We also provide an example illustrating the idea we used in the proof in Figure 3 and lines 895 to 904 in Appendix E.
>
> Finally, for the empirical performance of these different reverse methods, please refer to the ablation studies (Table 3 and 5) in the paper.

---

> ### Author Response · Authors · 2024-11-20
> **Response to Reviewer iCzn (Part 2/2)**
>
> > **Q3:** Note that $S_n\subseteq [n]^n$. If the input data comprises of only permutations, then the network should learn to sample from a distribution whose samples are permutations and the standard framework of diffusion models applies.
> >
> >
> > The authors also mention that representing a transition matrix over $S_n$ requires $n!\times n!$ sized matrix. However, the authors themselves give a succinct description/representation of the forward transition matrix in the paper. The authors should elaborate why it is not possible to use this representation algorithmically.
> >
>
> **A3:** It is true that we can view each permutation in $S_n$ as a sequence in $[n]^n$. However, standard diffusion models assume token-wise conditional independence when modeling the reverse transition distributions. This assumption does not hold in learning permutations since different components of $X_{t-1}$ are **not** **independent** conditioned on $X_t$ in the reverse process, and they have to satisfy the constraint of permutations (i.e., being one of the vertices of Birkhoff Polytope). Therefore, if we have a distribution over $[n]^n$, the denoising step in standard diffusion models would lead to noisy data $X_{t-1}$ that is not an exact permutation. Furthermore, it is also computationally expensive to project it to a distribution over $S_n$.
>
> Discrete diffusion methods like D3PM [4], which model categorical distributions, are also unsuitable for $S_n$. These methods require explicit matrix multiplications involving $n!\times n!$ transition matrices. While D3PM uses dense transition matrices such as uniform or discretized Gaussian distributions, performing dense matrix multiplications at this scale is impractical.
>
> Our proposed method addresses these challenges by defining efficient, customized transition distributions through card-shuffling methods. This approach avoids explicit matrix multiplications by directly simulating the forward process using the efficient operations of shuffling methods. Essentially, the shuffling methods induce “sparse” transitions on $S_n$, resolving the efficiency issues inherent in existing discrete diffusion models. As our framework is fundamentally different and existing frameworks are infeasible for $S_n$, our baselines focus on comparing different shuffling methods within our framework.
>
> Thanks for pointing out the related references [2,3]. We have acknowledged their contributions in the updated paper **(lines 84–86 on page 2, marked in blue)**. Work [2] extends D3PM with a continuous-time Markov chain approach but still involves costly computations tied to the transition distribution, making it challenging to apply directly to $S_n$. Work [3] models a diffusion process on sequences in $\mathcal{X}^L$, changing one index of the sequence at each forward step.
>
> Applying [3] to $S_n$ would involve modeling sequences in $[n]^n$ using [3], with $X_0$ as the data distribution of the permutations. While Glauber dynamics in [3] avoids the conditional independence issue mentioned earlier, one caveat is that $X_t$ for $t\geq 1$ would most likely lie outside $S_n$.  In the reverse process, if learning is imperfect, the final sampled sequence may not be a permutation, necessitating projection onto $S_n$, which is again a non-trivial task. Currently, the code for [3] is not publicly available. We find their approach intriguing and look forward to experimenting with GGM once the code is released.
>
> > **Q4:** In proposition 1, should it be changed to "the GPL distribution can represent a delta distributions in the limit" instead of "exactly"?
> >
>
> **A4:** You are correct since we are using $-\infty$ for the logits. We have reorganized the expressiveness results in the newly uploaded version of the paper. The original Proposition 1 is now separated into Proposition 1 **(colored blue)** in the main paper and Lemma 4 in Appendix E. The expressiveness result for GPL is stated as Theorem 2 **(colored blue)** in the main paper and proved in Appendix E.
>
> ## References
>
> [1] Generating a random permutation with random transpositions by Diaconis and Shahshahani
>
> [2] Simplified and Generalized Masked Diffusion for Discrete Data by Shi et al.
>
> [3] Glauber Generative Model: Discrete Diffusion Models via Binary Classification by Varma et al.
>
> [4] Austin et al. "Structured denoising diffusion models in discrete state-spaces." Advances in Neural Information Processing Systems 34 (2021): 17981-17993.

---

> ### Comment · Reviewer_iCzn · 2024-11-24
>
> Thank you for the response. I appreciate the proof regarding the GPL distribution and regarding the scale of the experiments. However, I have some further concerns with response A3. The authors quote that the original D3PM work assumed factored denoising and thus the denoising trajectory cannot be ensured to be a permutation.
>
> 1. While this is true, there are many other proposals for diffusion models which do not assume factorization. For instance SEDD ([1a]) does not assume factorization of the reverse distribution and would be a valid baseline for the current work.
>
> 2. When modeling distributions over $[n]^n$ it is not necessary that the entire trajectory is restricted to $S_n$. It is sufficient if the algorithm outputs a permutation at time $0$.
>
> 3. I also want to point out that D3PM style models have been shown to be capable of planning (such as generating SuDoKus, Solving SAT problems), where factorization certainly does not hold and there are lots of structure and constraints in the output. I refer to [2a], [3a] and references in the paper. Even language modeling and image generation involve structure and constraints, where factorization does not hold, yet D3PM based models have been successful.
>
> 4. Regarding Glauber Generative model [3], I see that their algorithm has been clearly described in the paper and can be implemented in a straightforward manner.
>
> I am not satisfied with the baselines used in this work and the absence of other discrete diffusion based approaches.
>
> [1a] Discrete Diffusion Modeling by Estimating the Ratios of the Data Distribution
>
> [2a] BEYOND AUTOREGRESSION: DISCRETE DIFFUSION FOR COMPLEX REASONING AND PLANNING
>
> [3a] LayoutDM: Discrete Diffusion Model for Controllable Layout Generation

---

> > ### Author Response · Authors · 2024-11-26
> > **Further Response Part 1/2: Experiments on ImageNet Jigsaw Puzzle and TSP-50**
> >
> > Thank you for your reply. We would first like to share some preliminary results on the ImageNet jigsaw puzzle and TSP-50 experiments. For the ImageNet jigsaw puzzle experiments, due to time constraints, we ran on only the first 20 classes of the ImageNet dataset, and we currently only have results for $2\times 2$ and $3\times 3$. In the table below, the first two rows are the results of our method, and the last two rows are results from the Gumbel-Sinkhorn network. It is clear that our model outperforms the Gumbel-Sinkhorn network.
> >
> > |  |  | **Kendall-Tau $\uparrow$** | **Accuracy (%)** | **Correct (%)** | **RMSE** $\downarrow$ | **MAE** $\downarrow$ |
> > | --- | --- | --- | --- | --- | --- | --- |
> > | **SymmetricDiffusers (Ours)** | **$\mathbf{2\times 2}$** | **0.8627** | **83.50** | **89.88** | 0.1806 | **0.0413** |
> > |  | **$\mathbf{3\times 3}$** | **0.7451** | **57.50** | **79.31** | **0.2245** | **0.0687** |
> > |  |  |  |  |  |  |  |
> > | **Gumbel-Sinkhorn Network** | **$\mathbf{2\times 2}$** | 0.8212 | 78.36 | 86.26 | **0.1583** | 0.0687 |
> > |  | **$\mathbf{3\times 3}$** | 0.5667 | 19.10 | 55.83 | 0.3388 | 0.1636 |
> >
> > For the TSP-50 experiment, Concorde and 2-Opt are OR solvers, and GCN, DIFUSCO, and Ours are learning-based models. Due to time constraints, we trained on only 1/3 of the training set (500K graphs), and the numbers for 2-Opt, GCN, and DIFUSCO are directly copied from the DIFUSCO paper [1]. The decoding heuristics used for GCN and DIFUSCO are greedy. The decoding heuristics used for our method are beam search and picking the tour with the shortest length in the final beam. Although our current results do not surpass the state-of-the-art, they are still comparable and demonstrate significant promise. As our method is not specifically tailored for TSPs, further hyperparameter tuning and architectural tweaking are required. Importantly, such modifications would not impact our core contribution, i.e., the discrete diffusion framework over finite symmetric groups. We plan to continue exploring this direction and will provide updated performance results on large-scale TSPs in a future version of our paper.
> >
> > | **Method** | **Concorde** | **2-Opt** | **GCN** | **DIFUSCO** | **Ours** |
> > | --- | --- | --- | --- | --- | --- |
> > | **Tour Length $\downarrow$** | **5.69** | 5.86 | 5.87 | **5.70** | 5.86 |
> > | **Optimality Gap (%) $\downarrow$** | **0.00** | 2.95 | 3.10 | **0.10** | 2.94 |
> >
> >
> > ### References
> >
> > [1] Zhiqing Sun and Yiming Yang. Difusco: Graph-based diffusion solvers for combinatorial optimization, 2023.

---

> > > ### Author Response · Authors · 2024-11-26
> > > **Further Response Part 2/2: Other Discrete Diffusion Baselines**
> > >
> > > We then address your concerns on other discrete diffusion baselines. For all of the work you referenced [2, 3, 4, 5], whether or not they assume conditional independence or factorization, the diffusion trajectory would not be restricted to $S_n$ in our problems. You are correct that as long as the algorithm outputs a permutation at time 0, then it is not necessary for the entire trajectory to be restricted to $S_n$. However, if we do not impose the constraint of having the trajectory to be in $S_n$ in the design of the algorithm, there is a high chance that the final sample of the algorithm would not be a permutation, especially for large scale problems. In [4], although tasks such as solving Sudokus or SAT problems indeed has lots of structures and constraints in their outputs, the search spaces of these problems are not significantly large. For the Sudoku task, we manually inspected their dataset and found that in the inputs, there are on average 30 initially filled cells out of the 81 cells. So the search space for Sudoku has size around $9^{50}<40!$, and our model considered search spaces with sizes up to $|S_{200}|=200!$ in the experiments. For the SAT problem, [2] considered 9 variables at most with $2^9=512$ search space size, which is also significantly smaller than the search space we considered in our paper. Intuitively, the combinatorial structure (e.g., permutation in our case) of the problem should only help us solve the problem. If we disregard the underlying structure in our models, it would be hard to scale up.
> > >
> > > For large-scale permutation learning problems, if the final sample is not a permutation, then we need to perform a projection onto $S_n$. This is a non-trivial convex optimization task which requires iterative optimization solvers. This additional projection step not only slows down the sampling process, but the distribution over $S_n$ after the projection is also not guaranteed to be correct.
> > >
> > > Nevertheless, we are glad to experiment with the baselines you proposed. Due to time constraints, we are currently focusing on adapting Score Entropy [3] to our experiment setup, and we will post a follow-up response if our experiments finish before the discussion ends.
> > >
> > > Finally, **we note that [2] and [4] are also concurrently under review in ICLR 2025 (submission number 9621 and 4441, respectively)**, so we kindly point out that it is unfair to request a comparison with them.
> > >
> > >
> > > ### References
> > >
> > > [2] Glauber Generative Model: Discrete Diffusion Models via Binary Classification by Varma et al.
> > >
> > > [3] Discrete Diffusion Modeling by Estimating the Ratios of the Data Distribution
> > >
> > > [4] BEYOND AUTOREGRESSION: DISCRETE DIFFUSION FOR COMPLEX REASONING AND PLANNING
> > >
> > > [5] LayoutDM: Discrete Diffusion Model for Controllable Layout Generation

---

> > > > ### Comment · Reviewer_iCzn · 2024-11-27
> > > >
> > > > Thank you for the prompt response. I understand your claim that an algorithm which works on $S_n$ naturally can be expected to be better than off the shelf algorithms which can work on the product space. However, my point was that this fact has to be established with rigorous experiments -- since this is one of the fundamental motivations behind the present work.
> > > >
> > > > Regarding scale: SEDD (and PLAID, MD4, GGM etc ) works on language modeling tasks with a sequence length of ~1000 and a vocabulary length of ~30000 and is certainly not a product distribution (there are several places where grammar imposes strict constraints and structure). Therefore, one could imagine them performing well in planning tasks as well. My point with citing several recent works (and older ones) was that regular diffusion models have been used in planning tasks quite successfully and therefore make for a strong baseline (whether SEDD or GGM or D3PM based works).
> > > >
> > > > Including at least one or two of those would strengthen your argument. Looking forward to the results (if ready by then). I will discuss further with the AC and other reviewers based on this. I want to stress that I like this line of work and find it very promising.

---

> ### Author Response · Authors · 2024-12-03
> **Experiments Follow-up**
>
> Thank you for your response, and we definitely agree that intuitions should be backed up by experiments. We have tested our model against SEDD, which is currently one of the strongest discrete diffusion models. To clearly compare our method and SEDD, we set up a simple experiment for the models to learn a delta distribution over $S_n$ (i.e. a single permutation) for $n=100$ and $200$. In particular, we let the models learn the identity permutation and a fixed arbitrary distribution.
>
> For SEDD, we view a permutation as a sequence of length $n$, where each number of the sequence is from $\\{0,1,\ldots,n-1\\}$. While the sequence at time 0 is a permutation, the trajectory may fall outside of $S_n$ for SEDD. We use the uniform transition for the forward process following the original work. For the reverse process of SEDD, we start by sampling a random sequence of length $n$.
>
> For our method, we use riffle shuffle as the forward process and the GPL distribution as the reverse process. The entire trajectory is restricted in $S_n$. At the start of the reverse process, we sample a permutation from $S_n$ uniformly at random.
>
> The SEDD model we use in our experiments has about 25M parameters, while our model only has about 2M parameters. For $n=100$, we use a batch size of 512 and 30K training steps on both methods. For $n=200$, we use a batch size of 128 and 30K training steps on both methods. For performance evaluation, we randomly sample 2560 sequences for SEDD and 2560 permutations for our method, and we perform their respective decoding processes.
>
> The results are detailed in the following three tables with experiment setting stated in the header.
>
> | **The Identity Permutation, $n=100$** | **Accuracy (%)** | **Correct (%)** |
> | --- | --- | --- |
> | **SymmetricDiffuser (Ours)** | **100** | **100** |
> | **SEDD** | 95.47 | 99.95 |
>
> | **Fixed Arbitrary Permutation,** $n=100$  | **Accuracy (%)** | **Correct (%)** |
> | --- | --- | --- |
> | **SymmetricDiffuser (Ours)** | **100** | **100** |
> | **SEDD** | 93.24 | 99.93 |
>
> | **Fixed Arbitrary Permutation,** $n=200$ | **Accuracy (%)** | **Correct (%)** |
> | --- | --- | --- |
> | **SymmetricDiffuser (Ours)** | **100** | **100** |
> | **SEDD** | 88.75 | 99.94 |
>
> Our method reaches 100% accuracy in all experiments. While the accuracies of SEDD are also high, there are still notable gaps between SEDD and our method, particularly as the sequence length increases. We also observe that SEDD makes mistakes exactly because some of the samples are not permutations. These experiments demonstrate that by restricting the trajectory to $S_n$ and leveraging the structures of $S_n$, our method is more effective than previous discrete diffusion models that rely on transitions in the larger sequence spaces. We plan to include these experiments in a future version of our paper. We also plan to conduct a more comprehensive analysis in the future, including tasks like learning mixture distributions.
>
> We would also like to highlight that it is **nearly impossible** for other discrete diffusion models (including SEDD) to solve the tasks introduced in our paper, including the jigsaw puzzle, sorting multi-digit MNIST numbers, and the TSP. The reason is that all prior discrete diffusion models assume a **fixed** alphabet or vocabulary, and they model categorical distributions on the fixed alphabet. For example, in NLP tasks, the vocabulary is predefined and fixed. However, the alphabet is the set of all possible image patches for image tasks such as jigsaw puzzles and sorting MNIST numbers. It is impractical to gather the complete alphabet beforehand. We could potentially train VQVAEs to obtain quantized image embeddings. However, such an approach introduces the approximation error from the quantized alphabet. For the TSP, each node of the graph is a point in the continuous space $\mathbb{R}^2$, so it is also impossible to gather the complete alphabet. In contrast, our method can be successfully applied to these tasks because we model a distribution on the fixed alphabet $S_n$, and we treat each permutation as a function that can be applied to an ordered list of objects.
>
> Finally, we apologize for the delay in our response, as it took us additional time to modify the code of SEDD. We hope that our response addresses your concerns thoroughly.

---

> > ### Comment · Reviewer_iCzn · 2024-12-03
> >
> > Thank you very much. This addresses all my concerns. I will increase the score to 8.

---

### Official Review · Reviewer_nrB8 · 2024-11-01

**Soundness:** 4
**Presentation:** 4
**Contribution:** 4
**Rating:** 8
**Confidence:** 2

**Summary:**

This paper explores a novel problem: learning probability distributions over finite symmetric groups. As an example, the problem can be thought of as learning to sort, with important differences:

- instead of finding just the most optimal permutation, the focus is on learning a *distribution* over all possible rankings, so that rankings "closer" to the optimal permutation are more likely to be generated.
- the problem formulation is general enough to cover rankings over any finite group of elements and not just a set of $n$ integers as long as one trains the algorithm using appropriate data (for instance, pictures of numbers instead of just numbers).

The authors propose *SymmetricDiffusers*, a discrete diffusion model (with a transformer-based architecture in this case) trained to recover the target permutation in several steps after the original (target permutation) is converted into noise over a number of steps. This decomposition of the recovery process eases the otherwise very difficult problem of learning to directly come up with the optimal permutation out of $n!$ possible states.

With this context, the authors' study reveals several insights into potentially best practices with regard to the training process, *e.g.,* choice of the forward shuffling method (riffle shuffle because of its fast mixing time), and when it might be feasible to merge steps during the denoising process. The authors validate their findings via state-of-the-art results on three benchmarks: MNIST sorting, jigsaw puzzles, and the Travelling Salesman Problem.

**Strengths:**

1. The style of writing is extremely clear and structured.
2. The paper makes a wealth of interesting contributions: introducing a novel perspective on an important problem that is currently underexplored in the ML community; proposing a method to solve said problem; revealing insights that will help further research; and finally, validating their method on three benchmarks. The authors also share their code. The general nature of the problem means that the findings here can potentially open up a whole new set of possibilities.

**Weaknesses:**

I don't see obvious weaknesses. But, as the authors also acknowledge (in the conclusion and Appendix G), there is potential for improving scalability (w.r.t. $n$) and possibly extending the method to finite groups beyond $S_n$.

**(Nit: typos)**
- Abstract (line 009): groups --> group?
- line 472: performances --> performance?

**Questions:**

1. Have the authors considered evaluating OOD performance (e.g., feeding colored, or otherwise font-shifted MNIST into a model that was trained on grayscale images) ? Do they anticipate a drop in performance in that setting?

---

> ### Author Response · Authors · 2024-11-20
> **Response to Reviewer nrB8**
>
> Thank you for the insightful and constructive comments. We appreciate your positive feedback! We have corrected the typos in the newly uploaded version of the paper.
>
> > **Q1:** Have the authors considered evaluating OOD performance (e.g., feeding colored, or otherwise font-shifted MNIST into a model that was trained on grayscale images)? Do they anticipate a drop in performance in that setting?
> >
>
> **A1:** Evaluating the OOD performance of our model is indeed an interesting and important direction, and we would anticipate a performance drop in such settings. Feeding colored or font-shifted MNIST into a model that was trained on grayscale images would primarily test the OOD performance of the underlying vision encoder (e.g., the CNN) of SymmetricDiffuser. Testing the generalization ability with varying sequence lengths would also be interesting. For example, we could train on short-length sequences and test on long sequences. However, the focus of this paper is on proposing a novel discrete diffusion framework for symmetric groups, and enhancing OOD performance falls outside the scope of our current study. Addressing OOD performance would likely require significant modifications to the loss function or model design, making it a promising topic for future research.

---

> > ### Comment · Reviewer_nrB8 · 2024-11-26
> >
> > Thank you for your response. Indeed, OOD performance can be taken up as a separate research topic.
> >
> > As I did not have any major concerns to begin with, I intend to maintain my current score of 8.
> >
> > Congratulations on your great work!

---

### Official Review · Reviewer_7CmY · 2024-11-08

**Soundness:** 4
**Presentation:** 4
**Contribution:** 3
**Rating:** 8
**Confidence:** 3

**Summary:**

This paper presents a discrete diffusion model for learning a distribution over permutations ($S_n$). They present several choices of forward noising process, including transpositions, insertions, and the rifle shuffle. For the reverse noising process, they define inverse processes corresponding to each of the three proposed forward processes, as well as a generalization of the Plackett-Luce Distribution that is more expressive than the original (e.g. it can represent a delta distribution). They train their diffusion models via a variational lower bound, estimated via Monte Carlo samples since they cannot obtain an analytic form. They derive a noise schedule, based on merging adjacent diffusion steps for certain inverse distributions, and run experiments comparing different versions of their diffusion model to differentiable sorting methods and test tasks including sorting MNIST digits and solving traveling salesman instances.

**Strengths:**

This is the first work to define a diffusion model over $S_n$. Their enumeration, and mathematical treatment, of possible forward and reverse noising processes is thorough, as are the experiments and ablations. The presentation is quite clear in general. The application of a diffusion model to problems that require outputting a permutation in a differentiable matter — even if a distribution over $S_n$ is not strictly required — seems to be creative and novel. The experimental results are generally good, particularly at increased sequence lengths.

**Weaknesses:**

To me, the main piece missing from this paper is a motivation of why machine learning applications require generative models over permutations in the first place, rather than just outputting a single permutation, which is what the tested datasets seem to satisfy. Is this related to differentiability? The introduction is well-written, but would improve from being more concrete and grounded in the machine learning literature. To be specific, the claim on line 35 that “Therefore, studying probabilistic models over $S_n$ through the lens of modern machine learning is both natural and beneficial” feels a bit unjustified. Differentiable sorting is mentioned in the related works, but a discussion of why “such methods often focus on finding the optimal permutation instead of learning a distribution over the finite symmetric group” and what the tradeoffs are would be helpful.

The experiments also show stronger performance for longer sequence lengths, but the quadratic scaling with longer sequence lengths remains an open direction of improvement.

**Questions:**

1. As discussed in the Weaknesses section, what exactly is the motivation of using diffusion model for these problems, if ultimately only a single learned permutation is required per input? This is my primary question.
2. For the right choice of parameters, can the reverse processes actually represent the exactly correct distributions induced by the corresponding forward diffusion process?
3. One minor point of confusion was that the abstract claims to learn a distribution over $S_n$, but the concrete objects that are dealt with are ordered sets of objects (stored in an $n$ by $d$ matrix). Would it be accurate to refer to this method as *conditional* diffusion? If not, how could the architectures best be modified to output a distribution over raw permutation matrices?
4. As noted on line 154, $\mathcal{S}$ does not change across steps — why enforce this for diffusion models? Does this make something easier? Is it potentially restrictive in terms of what distributions can be represented after a given number of steps?

---

> ### Author Response · Authors · 2024-11-20
> **Response to Reviewer 7CmY**
>
> Thank you for the insightful and constructive comments. We appreciate your positive feedback and address the questions below.
>
> > **Q1:** Why machine learning applications require generative models over permutations in the first place, rather than just outputting a single permutation, if ultimately only a single learned permutation is required per input?
> >
>
> **A1:** As in many other prediction tasks, predicting a distribution is often more useful than predicting a single output. In particular, in our context, there are several advantages.
>
> First, there are many tasks where the optimal permutation is not unique. For example, in the jigsaw puzzle experiments, there may be identical patches in an image, in which case there would be multiple permutations that can recover the image. We added noise to the MNIST dataset to disambiguate the patches purely to simplify the evaluation metric computations, and our framework is certainly able to solve jigsaw puzzles with identical patches. Another example is the TSP, where multiple permutations may exist that result in the same minimum tour length.
>
> Second, many NP-hard problems exist, e.g., TSPs and graph isomorphism problems, where the optimal solution is hard to find. Having a distribution over $S_n$ is extremely useful in constructing probabilistic approximate algorithms, e.g., MCMC-based search methods.
>
> Last but not least, learning a distribution over $S_n$ allows learning to sample permutations, which further enables learning to generate other discrete objects. For instance, when generating expander graphs, one of the key steps in the probabilistic construction is to generate random permutations of vertices, cf., [1].
>
> [1] Friedman, J., 2003, June. A proof of Alon's second eigenvalue conjecture. In Proceedings of the thirty-fifth annual ACM symposium on Theory of computing (pp. 720-724).
>
> > **Q2:** For the right choice of parameters, can the reverse processes actually represent the exactly correct distributions induced by the corresponding forward diffusion process?
> >
>
> **A2:** Yes! During the rebuttal period, we proved that the reverse process using GPL can represent **any** distribution over $S_n$, which is a significant result regarding the expressiveness of the reverse process. Please refer to Theorem 2 **(colored blue)** and Appendix E of the newly uploaded paper. We also provide an example illustrating the idea we used in the proof in Figure 3 and lines 895 to 904 in Appendix E.
>
> > **Q3:** The abstract claims to learn a distribution over $S_n$, but the concrete objects that are dealt with are ordered sets of objects (stored in an $n$ by $d$ matrix). Would it be accurate to refer to this method as *conditional* diffusion? If not, how could the architectures best be modified to output a distribution over raw permutation matrices?
> >
>
> **A3:** For a set of distinct objects/components, there is a one-to-one correspondence between all ordered sequences of the components and $S_n$. If there are identical components, then we can apply arbitrary tie-breaking rules and still obtain the one-to-one correspondence. Therefore, dealing with concrete objects is equivalent to learning a distribution over $S_n$. In our experiments, we are using conditional diffusions in the sense that we are conditioning on the set (i.e. the unordered collection) of components. We do not need any modifications to output a distribution over the raw permutations because of this one-to-one correspondence.
>
> > **Q4:** As noted on line 154, $\mathcal{S}$ does not change across steps — why enforce this for diffusion models? Does this make something easier? Is it potentially restrictive in terms of what distributions can be represented after a given number of steps?
> >
>
> **A4:** We let $\mathcal{S}$ to remain the same across steps to construct a time-homogeneous Markov chain. The first reason is that existing random walk theories on finite groups primarily deal with time-homogeneous Markov chains. If we allow $\mathcal{S}$ to change across steps, the convergence analysis may become very difficult in general. The second reason is that having $\mathcal{S}$ to change across time-steps may not bring any additional benefits. For example, we could mix riffle shuffles, random transpositions, and random insertions together in the forward process. However, we know that riffle shuffles mix the fastest, so having other shuffling methods will only slow down the mixing time. Experiments also show that the riffle shuffle alone is very effective with fast mixing time.

---

> > ### Comment · Reviewer_7CmY · 2024-11-27
> > **Thanks for the response**
> >
> > Thanks to the authors for the response. I found it clear, and maintain my accept rating. For whatever it's worth, I'd also encourage the authors to include more of A1 in the manuscript, or at least in any future poster or presentation, as it creates a stronger narrative background for their technical contributions.

---

### Official Review · Reviewer_H6zv · 2024-11-08

**Soundness:** 3
**Presentation:** 3
**Contribution:** 3
**Rating:** 8
**Confidence:** 3

**Summary:**

The paper presents SymmetricDiffusers, a novel discrete diffusion model for learning distributions over permutations within symmetric groups. This model addresses the complexity of directly modeling the vast and discrete state space of permutations by decomposing the task into simpler, more manageable transitions.

Key contributions include:
1) Forward Diffusion Process Using Card Shuffling Methods: Symmetric diffusers introduce noise to permutations using classical shuffling methods (riffle shuffles, random transpositions, and random insertions). These methods facilitate a gradual transformation toward a known noise distribution, simplifying the learning process.
2) Generalized Plackett-Luce (PL) Distribution for Reverse Diffusion: To return the noisy state to its original distribution, the model leverages a neural network-based generalized PL distribution, enhancing expressiveness and effectively reconstructing complex dependencies within permutations.
3) Theoretically Grounded Denoising Schedule: An optimized denoising schedule merges reverse steps to boost sampling efficiency and learning performance, reducing computational requirements without sacrificing accuracy.

The model demonstrates state-of-the-art or comparable results in tasks such as sorting, jigsaw puzzle assembly, and solving traveling salesman problems, validating its effectiveness in permutation-based applications.

**Strengths:**

Originality: This paper demonstrates notable originality by advancing discrete diffusion models to tackle the problem of learning distributions over permutations in symmetric groups. The task of modeling permutations is inherently challenging due to the factorial growth of the state space, and SymmetricDiffusers introduces an innovative solution by utilizing card shuffling techniques (riffle shuffle, random transpositions, and random insertions) as part of a structured forward diffusion process. Combining classic combinatorial methods with modern neural-based diffusion modeling is an inspired choice well-suited to the discrete and combinatorial nature of permutations. Compared to related works on discrete diffusion—such as Discrete Denoising Diffusion Probabilistic Models (D3PMs), which focus on multinomial categories—SymmetricDiffusers addresses a unique domain with a permutation-focused framework that current D3PMs do not target.

Furthermore, introducing a generalized Plackett-Luce (PL) distribution for the reverse process sets this work apart from other discrete diffusion models, such as score-based continuous-time discrete-state models, which operate on categorical data but lack this flexibility. The generalized PL distribution is well suited for structured dependencies in permutations, enabling greater expressiveness and more accurate learning over complex permutations.

Quality: The paper demonstrates high methodological rigor. Each component of SymmetricDiffusers—the forward process using card shuffling, the generalized PL distribution for reverse diffusion, and the denoising schedule—is technically well developed and grounded in established theories of random walks and Markov chains on finite groups. This adds a layer of mathematical credibility to the proposed model, particularly in the careful treatment of transition probabilities and mixing times associated with the shuffling methods.

Additionally, the experiments are comprehensive and well-chosen to validate the model across diverse tasks, such as sorting, jigsaw puzzle completion, and traveling salesman problems (TSPs). This diversity of functions effectively showcases the robustness and generalizability of SymmetricDiffusers, achieving state-of-the-art or comparable results in each case. Unlike other discrete diffusion models focused on simpler domains (e.g., categorical image data), SymmetricDiffusers’ application to more complex combinatorial problems uniquely highlights the model’s practical value and robustness.

Clarity: The paper is generally well-organized and easy to follow, with a clear introduction to the challenges and requirements of permutation modeling. Complex ideas are supported by notations, figures, and a well-structured flow that gradually builds the reader’s understanding from the background on symmetric groups to the technical construction of SymmetricDiffusers. Visualizing the forward and reverse processes in figure form is particularly helpful, illustrating the model’s approach to diffusion over permutations.
However, compared to some other papers on discrete diffusion (e.g., D3PMs), certain sections—particularly those explaining the forward and reverse processes in detailed mathematical terms—might benefit from additional simplification for readers less familiar with random walks on finite groups. Nevertheless, the overall clarity and structure make the contributions accessible and understandable.

Significance: SymmetricDiffusers addresses an important area in machine learning by advancing the state of permutation modeling within discrete diffusion frameworks. Distributions over permutations are critical in domains like ranking, combinatorial optimization, and sequence alignment, and current models struggle with the computational complexity posed by large discrete spaces. This work opens up new possibilities for generative modeling within these areas, especially with its robust performance in tasks requiring complex combinatorial reasoning (e.g., TSP).

Compared to recent discrete diffusion works on more general categorical or sequential data, this paper contributes uniquely to discrete generative modeling by directly addressing permutation structures. This is significant because it establishes a pathway for diffusion models to utilize high-complexity tasks beyond traditional applications effectively. By providing an effective means of modeling permutations, SymmetricDiffusers could substantially impact practical applications and inspire future research in discrete-state diffusion for structured data.

The paper’s strengths lie in its original approach, which combines structured combinatorial methods with diffusion modeling. It also has high-quality methodological rigor, clearly presents complex concepts, and significantly contributes to modeling distributions over large permutation spaces, positioning it as a valuable advancement in discrete diffusion research.

**Weaknesses:**

Limited Comparative Analysis with Other Discrete Diffusion Models: While SymmetricDiffusers clearly advances permutation modeling, the paper could benefit from a more detailed comparison with other discrete diffusion models, particularly those handling high-dimensional categorical or sequential data, such as Discrete Denoising Diffusion Probabilistic Models (D3PMs). The current related work section mentions other models briefly but does not delve into how SymmetricDiffusers directly compares in terms of handling large discrete spaces. Including more detailed experiments or ablation studies to demonstrate SymmetricDiffusers’ advantages over such models regarding efficiency or performance on simpler permutation tasks could clarify its relative strengths and limitations.

Lack of Exploration on Scalability to Larger: The factorial growth of permutations means that scaling to larger sizes can become computationally intensive. The paper currently demonstrates its model on small-to-moderate values (e.g., tasks like 4-digit MNIST sorting). However, it does not provide clear insights into the scalability limits or potential strategies for scaling SymmetricDiffusers to larger values, where the computational load may become a bottleneck. Future work should consider including benchmarks on larger values or discussing ways to optimize the model’s performance, such as using sparse transition matrices or adopting modular architectures.

Clarity in the Forward and Reverse Processes: While the paper is generally well-structured, some sections—particularly those detailing the forward and reverse diffusion processes—are highly technical and may be challenging for readers less familiar with symmetric groups and permutation modeling. Additional clarifications or simplified explanations could improve accessibility. Specifically, breaking down the mathematical formulations for each shuffling method and reverse diffusion process with more illustrative examples would make these sections easier to understand. Clearer definitions of key terms, such as “stationary distribution” and “mixing time” in the context of random walks, could also make the content more accessible to a broader audience.

Efficiency of the Denoising Schedule: The paper introduces a theoretically grounded denoising schedule to merge reverse steps and improve efficiency, but it lacks concrete benchmarks or ablation studies to assess its impact. Comparing SymmetricDiffusers with and without the denoising schedule in terms of computational time and performance would provide readers with a clearer understanding of its practical benefits. Additionally, exploring alternative denoising schedules or adaptive strategies that adjust based on task complexity could further optimize the model’s performance.

Broader Applicability and Practical Implications: Although SymmetricDiffusers demonstrates promising results in permutation-specific tasks (sorting, TSP, jigsaw puzzles), the paper could better communicate its broader applicability and potential limitations. For example, could the model be applied to non-permutation-based tasks with discrete structures, such as certain types of graph-based tasks? A brief discussion of the boundaries of SymmetricDiffusers’ applicability and how it might adapt to related yet distinct discrete structures would clarify its versatility and limitations.

Suggestions for Improvement:
1) Enhance comparative experiments by including Symmetric Diffusers alongside other discrete diffusion models (e.g., D3PMs) on more straightforward permutation tasks for more explicit benchmarks.
2) Expand on the model’s scalability for larger spaces, possibly with benchmarks on larger permutation spaces or theoretical discussions on extending to high-dimensional settings.
3) Provide additional clarifications and illustrative examples for complex forward and reverse diffusion processes sections.
Include an ablation study on the denoising schedule to quantify its impact on performance and efficiency.
4) Discuss the model's broader applicability to other discrete structures beyond permutations, providing readers with insight into its potential versatility.

**Questions:**

Comparative Analysis with Other Discrete Diffusion Models:
Question: Could you provide additional comparisons between symmetric diffusers and other discrete diffusion models, such as discrete denoising diffusion probabilistic models (D3PMs), particularly regarding tasks that involve smaller permutation spaces?
Suggestion: Including benchmarks with other models for more straightforward tasks could help clarify SymmetricDiffusers' unique benefits in terms of accuracy, expressiveness, and computational efficiency. If it outperforms or scales better than current discrete diffusion models, it would provide valuable evidence of its novelty.

Scalability for Larger :
Question: How do symmetric diffusers handle scalability as increases, and what strategies do you envision for managing the factorial growth in the state space for large permutations?
Suggestion: An empirical analysis or theoretical discussion on the scalability limits and potential optimizations (e.g., sparse transition matrices or modular architectures) would provide greater insight into the model’s practicality for large-scale tasks.

Denoising Schedule Efficiency:
Question: How significant is the denoising schedule's impact on computational efficiency and model performance? Have you considered alternative denoising schedules that might further improve efficiency?
Suggestion: An ablation study on the denoising schedule’s impact on performance and efficiency would be beneficial. Exploring adaptive denoising strategies or alternative merging techniques might also enhance the model’s flexibility and efficiency.

Forward and Reverse Diffusion Process Clarifications:
Question: Could you clarify certain technical details of the forward and reverse diffusion processes, especially for readers unfamiliar with symmetric groups? Specifically, could you break down the mathematical formulations for each shuffling method?
Suggestion: To make these sections more accessible to readers, illustrative examples for the shuffling and reverse diffusion steps or a more detailed explanation of terms such as “stationary distribution” and “mixing time” could be added.

Broader Applicability and Adaptation to Other Discrete Structures:
Question: Can symmetric diffusers be adapted for other structured discrete data beyond permutations, such as specific types of graph structures or other combinatorial tasks?
Suggestion: A brief discussion of the model’s applicability to other structured discrete data or potential adaptations would help clarify its versatility. For instance, could SymmetricDiffusers’ approach be generalized to data with hierarchical or relational structures?

Generalized Plackett-Luce Distribution Benefits:
Question: Could you provide more insights or experiments to demonstrate the practical benefits of using the generalized Plackett-Luce (PL) distribution over the standard PL model? In which cases does the generalized PL significantly enhance performance?
Suggestion: Examples or comparisons that highlight the generalized PL distribution’s expressiveness—especially in complex permutation tasks—would illustrate its value over standard PL models.

---

> ### Author Response · Authors · 2024-11-20
> **Response to Reviewer H6zv (Part 1/2)**
>
> Thank you for the insightful and constructive comments. We appreciate your positive feedback and address the questions below.
>
> > **Q1:** Could you provide additional comparisons between symmetric diffusers and other discrete diffusion models, such as discrete denoising diffusion probabilistic models (D3PMs), particularly regarding tasks that involve smaller permutation spaces?
> >
>
> **A1:** Existing discrete diffusion models assume token-wise conditional independence when modeling the reverse transition distributions. This assumption does not hold in learning permutations since different components of $X_{t-1}$ are **not** **independent** conditioned on $X_t$ in the reverse process, and they have to satisfy the constraint of permutations (i.e., being one of the vertices of Birkhoff Polytope). Therefore, if we have a distribution over $[n]^n$, the denoising step in standard diffusion models would lead to noisy data $X_{t-1}$ that is not an exact permutation. Furthermore, it is also computationally expensive to project it to a distribution over $S_n$.
>
> Discrete diffusion methods like D3PM, which model categorical distributions, are also unsuitable for $S_n$. These methods require explicit matrix multiplications involving $n!\times n!$ transition matrices. While D3PM uses dense transition matrices such as uniform or discretized Gaussian distributions, performing dense matrix multiplications at this scale is impractical.
>
> Our proposed method addresses these challenges by defining efficient, customized transition distributions through card-shuffling methods. This approach avoids explicit matrix multiplications by directly simulating the forward process using the efficient operations of shuffling methods. Essentially, the shuffling methods induce “sparse” transitions on $S_n$, resolving the efficiency issues inherent in existing discrete diffusion models. As our framework is fundamentally different and existing frameworks are infeasible for $S_n$, our baselines focus on comparing different shuffling methods within our framework.
>
> > **Q2:** How do symmetric diffusers handle scalability as increases, and what strategies do you envision for managing the factorial growth in the state space for large permutations? An empirical analysis or theoretical discussion on the scalability limits and potential optimizations (e.g., sparse transition matrices or modular architectures) would provide greater insight into the model’s practicality for large-scale tasks.
> >
>
> **A2:** First, our method significantly improves the scalability of existing approaches. As evidenced by prior work, scalability remains a highly challenging problem in permutation learning. For instance, in the 4-digit MNIST sorting experiment, baseline methods are only effective for sequence lengths up to 32. In contrast, our method achieves promising results for lengths up to 200, outperforming these baselines by a large margin.
>
> Second, as mentioned in A1, shuffling methods serve as efficient "sparsification" of D3PM transition matrices, which is crucial to our scalability improvement. While our framework has achieved significant progress, there is definitely lots of room (e.g., better parameterization of forward/reverse diffusion steps, better neural network architecture, and so on) for further scalability improvement, and it would be an exciting topic for future research.
>
> > **Q3:** How significant is the denoising schedule's impact on computational efficiency and model performance? Have you considered alternative denoising schedules that might further improve efficiency?
> >
>
> **A3:** The denoising schedule is an important hyperparameter in our model. We have already conducted an ablation study on the denoising schedule and provided a detailed discussion in Appendix G.2 and Table 6. We have also provided a theoretical justification and empirical guidelines for choosing the denoising schedule in Section 3.4 of the main paper.
>
> > **Q4:** Could you clarify certain technical details of the forward and reverse diffusion processes, especially for readers unfamiliar with symmetric groups? … To make these sections more accessible to readers, illustrative examples for the shuffling and reverse diffusion steps or a more detailed explanation of terms such as “stationary distribution” and “mixing time” could be added.
> >
>
> **A4:** Thank you for the suggestion. We will include additional explanations and illustrative examples of these concepts in the final version of the paper to improve accessibility.

---

> ### Author Response · Authors · 2024-11-20
> **Response to Reviewer H6zv (Part 2/2)**
>
> > **Q5:** Can symmetric diffusers be adapted for other structured discrete data beyond permutations, such as specific types of graph structures or other combinatorial tasks?
> >
>
> **A5:** Our model is directly applicable if a combinatorial problem with other structured discrete data can be equivalently reformulated using permutations. For instance, permutations can be used to represent solutions to TSP on graphs and exact graph matching (or graph isomorphism testing). For combinatorial problems that can not be reformulated using permutations, permutations could still be instrumental, e.g., generating expander graphs and various assignment problems. Given the central role of permutations in various combinatorial problems, our model has the potential to address a wide range of tasks involving structured discrete data.
>
> > **Q6:** Could you provide more insights or experiments to demonstrate the practical benefits of using the generalized Plackett-Luce (PL) distribution over the standard PL model?
> >
>
> **A6:** We have conducted the ablation studies and experiments in Appendix G.2. As seen from the results, the GPL distribution has better performance when sorting 4-digit MNIST numbers with $n=52$ .
>
> From a theoretical perspective, the standard PL distribution cannot represent a delta distribution, which is the ground truth for many problems. During the rebuttal period, we further proved that the reverse process using GPL can represent **any** distribution over $S_n$, which is a significant result regarding the expressiveness of the reverse process. This result is formalized in Theorem 2 **(colored blue)** and proved in Appendix E of the newly uploaded paper. We also provide an example illustrating the idea we used in the proof in Figure 3 and lines 895 to 904 in Appendix E.

---

### Meta-Review · Area_Chair_3khD · 2024-12-19

**Metareview:**

This paper develop techniques to learn discrete diffusion models over the group of permutations.

The reviewers and I unanimously appreciated:
- The novelty of the approach: Learning a diffusion model to sample on the set of permutation.
- The clarity of the writing.
- The quality of the contribution: the technical contributions are sound and relevant to the work.

The only weakness of the paper is that the scalability of the method is not fully explored (in particular for large values of n).

**Additional Comments On Reviewer Discussion:**

Reviewers unanimously agreed that this paper should be accepted.

I would not mind if this paper 'only' gets a spotlight or a poster but I am strongly recommending this paper to be accepted.

---

### Decision · Program_Chairs · 2025-01-22

Accept (Oral)